# GENERATIVE MODELING OF WEIGHTS: GENERALIZATION OR MEMORIZATION?

## ABSTRACT

Generative models, with their success in image and video generation, have recently been explored for synthesizing effective neural network weights. These approaches take trained neural network checkpoints as training data, and aim to generate high-performing neural network weights during inference. In this work, we examine four representative, well-known methods in this emerging area on their ability to generate *novel* model weights, *i.e.*, weights that are different from the checkpoints seen during training. Contrary to claims in prior work, we find that these methods synthesize weights largely by memorization: they produce either replicas, or at best simple interpolations, of the training checkpoints. Current methods fail to outperform simple baselines, such as adding noise to the weights or taking a simple weight ensemble, in obtaining different and simultaneously high-performing models. Our further results suggest that the memorization potentially resulted from limited data, overparameterized models, and the underuse of structural priors specific to weight data. Our findings highlight the need for more careful design and evaluation of generative models in new domains.

## 1 INTRODUCTION

Generative models, particularly diffusion models for image and video synthesis, have advanced significantly in recent years. Models such as Stable Diffusion (Rombach et al., 2022; Esser et al., 2024), Imagen (Ho et al., 2022), and FLUX (Black Forest Labs, 2024) demonstrate exceptional photorealism, with widespread applications in commercial art and graphics. Beyond static images, generative video models like Sora (Brooks et al., 2024) and Veo 3 (DeepMind, 2025) have recently gained attention, achieving impressive consistency and coherence in video synthesis. The success of these models is enabled by the strong priors for generative modeling from pre-trained representations (Esser et al., 2021; Radford et al., 2021; Yu et al., 2024) and the algorithmic designs tailored to the visual modalities (Johnson et al., 2016; Zhu et al., 2017; Peebles & Xie, 2023).

Building on this success, recent studies (Schürholt et al., 2022; Peebles et al., 2022; Erkoç et al., 2023; Wang et al., 2024) have extended the use of generative models to synthesize weights for neural networks. These methods collect network checkpoints trained with standard gradient-based optimization, and apply generative models to learn the weight distributions and produce new checkpoints, without direct access to the training data of the original task. The weights generated by these methods can often perform comparably to conventionally trained weights: they achieve high test accuracy in image classification models and high-quality 3D shape reconstructions in neural field models, across diverse datasets and model architectures.

In this study, we seek to answer an important question: have the generative models learned to produce meaningfully distinct weights that *generalize* beyond the training set of checkpoints, or do they merely *memorize* and reproduce the training data? While prior work has focused on evaluating these methods based on the performance of the generated models on the downstream tasks, this question is critical to understanding both the fundamental mechanisms and the practicality of these methods. To investigate this question of generalization, we analyze four representative weight generation methods (Schürholt et al., 2022; Peebles et al., 2022; Erkoç et al., 2023; Wang et al., 2024), covering different generative models and downstream tasks. These methods have been widely-studied (Dravid et al., 2024; Liang et al., 2024; Soro et al., 2024; Zhou et al., 2024; Cao et al., 2025; Wang et al., 2025; Zhang et al., 2025a;b), and claim to generate novel weights.

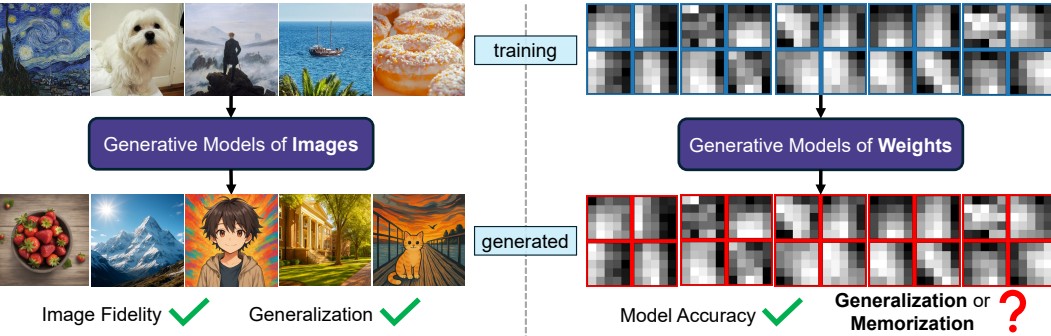

Figure 1: Building on their success in image generation, generative models have recently been applied to synthesize weights for neural networks. While they can produce effective neural network checkpoints (*e.g.*, classification models with high accuracy), it is unclear whether they can *generalize* beyond the training set to generate *novel* weights.

We first find the nearest training checkpoint to each generated checkpoint, to assess the novelty in the generated checkpoints. Contrary to these methods' claims, *almost all* generated checkpoints *closely resemble* specific training checkpoints in weight values, showing far less novelty than a new model trained from scratch. Beyond weight space similarity, we also examine the behaviors of generated models and their nearest training models. We compare the decision boundaries for classification models and the reconstructed 3D shapes for neural field models. In both cases, these generated models, which are very close to training models in weight space, also exhibit highly similar *outputs*.

Further, we show that current generative modeling methods offer no advantage over simple baselines for creating new model weights, in terms of producing models that differ from training checkpoints in behavior while maintaining model performance. These baselines generate new weights by adding Gaussian noise to training weights or interpolating between them. To quantify how novel a generated model's behavior is relative to the behaviors of the training models, we compute a similarity metric for models based on their overlap in prediction errors on the test set.

We find that limited data, overparameterized models, and the underuse of structural priors in weight data likely contribute to this memorization. First, we show that scaling up the training dataset can effectively reduce memorization without degrading the quality of the generated weights. Second, we demonstrate that the existing, highly over-parameterized weight generation models can memorize random weights, without learning meaningful patterns. Third, we find that currently used data augmentations are insufficient for generative models to learn the structural priors of weight data.

In summary, our findings consistently reveal *clear patterns of memorization in almost all generated checkpoints* from current methods, both in weight space and model behavior. We find that the generated weights largely replicate or interpolate the training weight data across all methods. As generative modeling continues to expand into new domains and modalities (Ravuri et al., 2021; Zrimec et al., 2022; Chi et al., 2023; Watson et al., 2023; Zeni et al., 2025), our findings highlight the importance of evaluating memorization in generated outputs, beyond standard quality metrics. More broadly, we hope this work can encourage researchers to consider both general properties of generative models and specific characteristics of each data modality in future methods.

## 2 BACKGROUND

This section provides an overview of the four generative models of weights we study, their differences from the hypernetwork methods, and the unique symmetries of neural network weight data.

### 2.1 GENERATIVE MODELING OF WEIGHTS

Generative models have recently been used to synthesize neural network weights, producing models that require no gradient-based optimization and perform comparably to models from standard training. In this study, we analyze four representative methods, spanning unconditional and conditional generation with autoencoders and diffusion models, under each method's primary experimental setup. We describe the primary setup of each method below, with more detail in Appendix A.

**Hyper-Representations** (Schürholt et al., 2021; 2022) generate neural network weights using an autoencoder. The autoencoder is trained on 2896 checkpoints of SVHN classification models with identical architectures but different initializations. After training, kernel density estimation (KDE) is applied to the latent representations of the top 30% checkpoints with highest accuracy. New weights are then generated by sampling a latent vector from the KDE-estimated distribution and decoding it.

**G.pt** (Peebles et al., 2022) is a conditional diffusion model that can generate new weights for a small predefined MNIST classification model architecture, given input weights and a target loss for the generated weights. It is trained on a collection of 2.1M model checkpoints from 10728 training runs, each paired with corresponding test losses. Once trained, G.pt can generate effective models by conditioning on randomly initialized weights and a minimal and fixed target loss (*e.g.*, 0).

**HyperDiffusion** (Erkoç et al., 2023) trains an unconditional diffusion model on 2749 neural field MLPs that represent 2749 unique 3D airplane shapes in ShapeNet (Chang et al., 2015). New shapes are generated by sampling a new set of MLP weights and reconstructing the mesh represented by it.

**P-diff** (Wang et al., 2024) trains an unconditional latent diffusion model on 300 neural network checkpoints. These checkpoints are saved at consecutive steps during an additional training epoch of the same base CIFAR-100 (Krizhevsky et al., 2009) classification model, after it has converged.

**Other methods**. Hypernetworks (Ha et al., 2016; Brock et al., 2018; Zhang et al., 2019; Knyazev et al., 2021; 2023) are neural networks trained to generate the weights of a target network, typically in a deterministic manner. Unlike the generative modeling methods that we study, hypernetworks are trained using supervision from downstream tasks rather than a collection of network checkpoints. As hypernetworks' generated weights often underperform compared to those obtained via gradient-based optimization, they are mainly used for weight initialization and neural architecture search.

## 2.2 Neural Network Symmetries

Neurons in a hidden layer have no inherent order, leading to permutation symmetry (Hecht-Nielsen, 1990) in neural networks: swapping neurons and adjusting weight matrices accordingly does not change a network's function. Another symmetry is scaling symmetry (Chen et al., 1993), including sign flips (multiplying all incoming and outgoing weights by -1) in `tanh` activations. Both G.pt and Hyper-Representations leverage permutation symmetry to augment weight data during training.

## 3 Evaluating Memorization in Weight Generation

To evaluate the novelty of generated model weights, we compare them to the original weights used to train the generative models of weight, analyzing both their weight values and model behaviors in comparison with various baselines.

## 3.1 Memorization in Weight Space

A natural first step in evaluating the novelty of generated weights is to find the nearest training weights to each generated checkpoint under $L_2$ distance, and check for replications in weight values. However, depending on the method, permutations of weight matrices in training checkpoints or autoencoder reconstructions of training weights must also be considered.

For methods (*e.g.*, G.pt) that apply weight permutation to augment data during training, we enumerate all possible permutations of training weights to identify the closest match for each generated checkpoint. Meanwhile, we find that Hyper-Representations' autoencoder cannot accurately reconstruct training weights, degrading accuracy, as shown in Figure 3. Thus, we compare its generated weights with the reconstructed training weights instead.

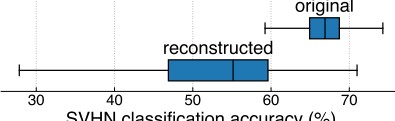

Figure 3: Reconstructing classification model weights with Hyper-Representations' autoencoder degrades model performance.

**Weight heatmap**. For each weight generation method, we visually inspect the three nearest training checkpoints for each of three randomly selected generated checkpoints using a heatmap of weights,

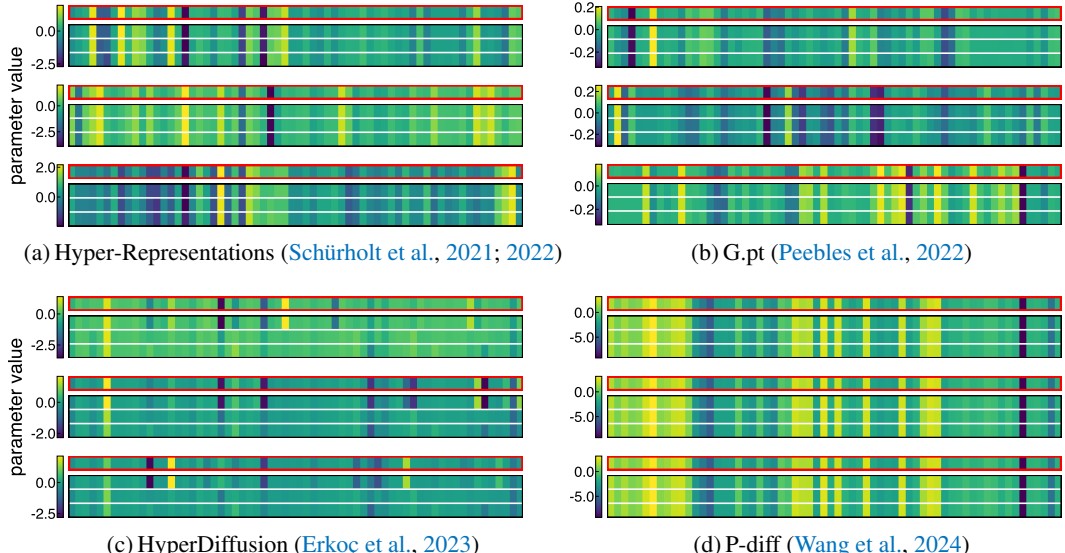

Figure 2: **Generated weights highly resemble training weights**. For each method, we display three heatmaps, showing weight values for 64 randomly selected parameter indices. In each heatmap, the top row (outlined in red) shows the values of a random generated checkpoint, and the three rows below (separated by white lines) show its three nearest training checkpoints. We observe that for every generated checkpoint, at least one training checkpoint is nearly identical to it.

shown in Figure 2 (more examples in Appendices C.1 and C.2). We observe that, for all sampled generated checkpoints across all methods, there is always at least one training checkpoint that closely resembles the generated one. Further, all of p-diff's training and generated checkpoints have nearly identical weight values, likely because its training checkpoints were saved consecutively from the same run, differing only by a small number of training updates.

**Distance to training weights**. In addition to visually inspecting weight values, we identify quantitative trends in weight value distributions that differentiate sampling a generated checkpoint from training a new model using standard gradient-based optimization (further results in Appendix B).

Specifically, we compute the $L_2$ distance from each training and generated checkpoint to its nearest training checkpoint (excluding self-comparisons for training checkpoints), and show the distance distributions in Figure 4. For all methods except p-diff, the generated checkpoints are significantly closer to the training checkpoints than training checkpoints are to one another. For instance, 94.4% of HyperDiffusion-generated checkpoints have an $L_2$ distance smaller than 10 to some training checkpoints, whereas any pair of training checkpoints has an $L_2$ distance above 50. This indicates that these methods produce models with lower novelty than training a new model from scratch. We note that the training checkpoints used in these methods are saved from *distinct* training runs.

For p-diff, we observed that the training checkpoints are much closer to each other than the generated checkpoints are to their nearest training checkpoints. However, the low distances between training checkpoints may be expected, since they are saved from the *same* training run at *consecutive* steps.

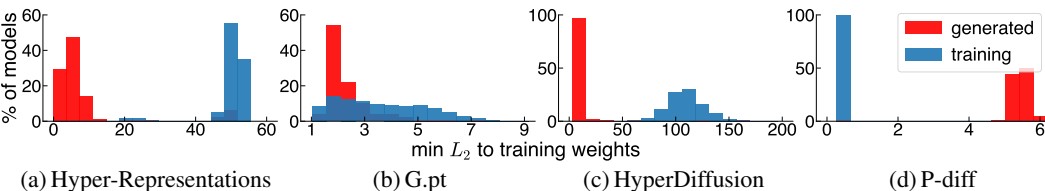

(a) Hyper-Representations     (b) G.pt     (c) HyperDiffusion     (d) P-diff

Figure 4: **Generated checkpoints are closer to training checkpoints than training checkpoints are to one another**, except for p-diff. This indicates that generated weights have lower novelty than a new model trained from scratch. The red and blue histograms represent the distributions of the $L_2$ distances to the nearest training checkpoints (excluding self-comparisons).

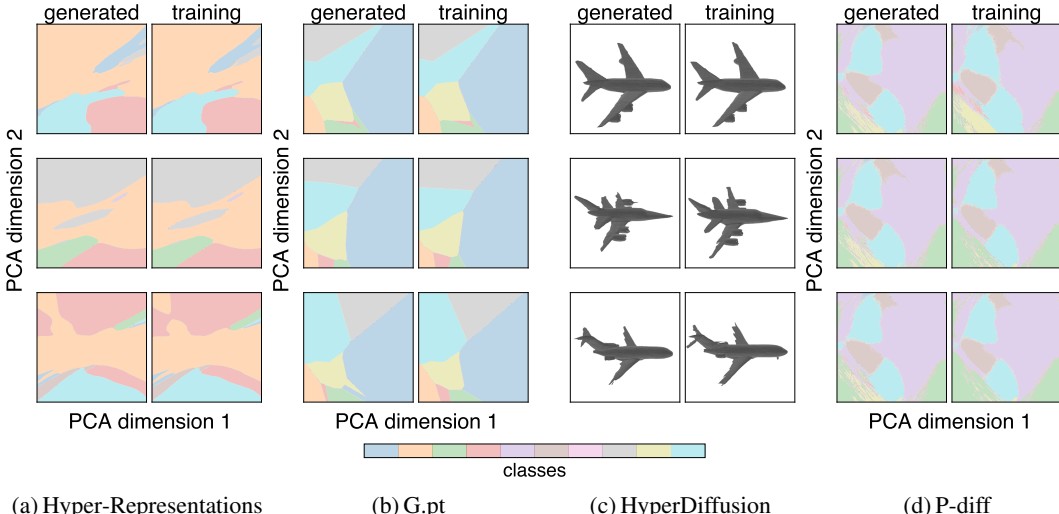

(a) Hyper-Representations  (b) G.pt  (c) HyperDiffusion  (d) P-diff

Figure 5: **Generated models produce highly similar outputs to their nearest training models**. Each row shows the decision boundaries or reconstructed 3D shapes of a randomly selected generated checkpoint ("generated") and its nearest training checkpoint ("training"). For p-diff models trained on CIFAR-100, decision boundaries are shown for ten randomly selected classes.

### 3.2 Memorization in Model Behaviors

In Section 3.1, we showed that generated weights highly resemble the training weights. However, similar weights can still yield different behaviors. Here, we compare the behaviors of generated models to the behaviors of their nearest training models in weight space. We also assess whether generative modeling methods differ from a simple noise-addition baseline for creating new weights.

**Model outputs**. To understand the behaviors of generated image classification models, we project each image dataset onto two principal components, and then visualize the models' decision boundaries. For HyperDiffusion, we reconstruct 3D shapes from the neural field models it generates.

For each method, we randomly select three generated checkpoints (additional examples in Appendices C.3 and C.4) and identify their nearest training checkpoints in weight space under $L_2$ distance, as in Section 3.1. Figure 5 presents the corresponding decision boundaries or 3D shapes. We find that generated models and their nearest training models produce highly similar predictions in image classification, as indicated by the nearly identical decision regions across all class labels. Similarly, the neural field models generated by HyperDiffusion also reconstruct to nearly identical 3D shapes as training ones, with visible differences only in minor details (*e.g.*, the edges of the airplane wings).

**Metric for novelty**. For generated weights to represent generalization, they need to behave sufficiently differently from training weights while maintaining high performance. To quantify the novelty of a generated classification model checkpoint, we adopt the model similarity metric from Wang et al. (2024), which measures the Intersection over Union (IoU) of incorrect test set predictions between two model checkpoints. The formal definition of this metric is in Appendix D.1. We explore an alternative similarity metric based on the percentage of prediction overlaps in Appendix D.3.

To assess a checkpoint's novelty, we compute its similarity with each training checkpoint and take the *maximum similarity*. A lower *maximum similarity* value indicates greater novelty, as it means the generated model's classification prediction error patterns differ more from all training models.

For HyperDiffusion, which generates neural field models rather than classifiers, we use Chamfer Distance (CD), a standard metric for 3D shapes. A lower minimum CD to the *test* shapes indicates better shape quality, analogous to higher classification accuracy. A higher minimum CD to the *training* shapes suggests greater novelty, akin to lower maximum similarity in classification models.

**Noise-addition baseline**. We compare the accuracy and maximum similarity of the generated checkpoints against a baseline that simply adds Gaussian noise to training weights. The weight generation methods are considered superior if, at the same level of novelty relative to training mod-

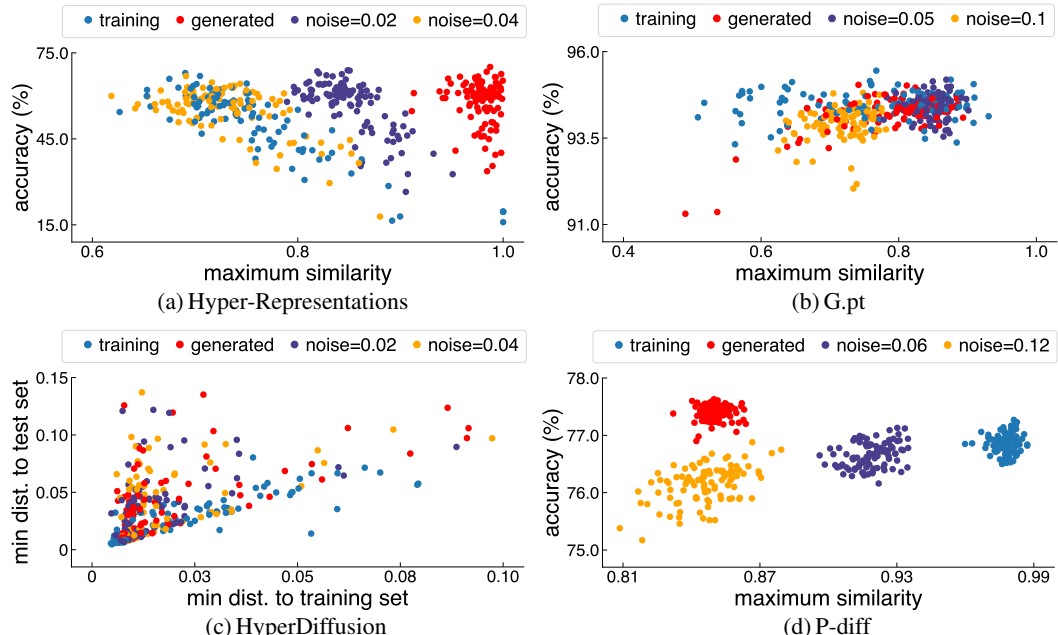

Figure 6: **Weight generation methods do not outperform noise addition in the accuracy-novelty trade-off**, except for p-diff. Novelty is measured by max error similarity (lower is better) or min point cloud distance to training checkpoints (higher is better). We show 100 samples per model type.

els, they produce models with better performances than noise addition. Figure 6 shows the accuracy and maximum similarity distributions for training models, generated models, and noise-added models. For each weight generation method, the noise amplitudes are chosen so that the maximum similarity of noise-added models roughly matches the maximum similarity of generated models.

**Accuracy-novelty trade-off**. As shown in Figure 6, for G.pt and Hyper-Representations, noise-added models often achieve comparable or even higher accuracy than generated models at the same maximum similarity to training models. Similarly, for HyperDiffusion, the distributions of the minimum CD to training and test shapes show no significant difference between generated and noise-added MLPs. These results suggest that the weight generation methods may *not* offer further benefits than simply adding noise to the training weights. An exception is p-diff, where generated models achieve a better trade-off between maximum similarity and accuracy than noise-added models.

### 3.3 P-DIFF GENERATES BY INTERPOLATION, NOT GENERALIZATION

As observed in Section 3.2, different from the other methods, p-diff achieves a better trade-off between the novelty and accuracy of generated models compared to the noise-addition baseline. Interestingly, the generated weights even surpass the training weights in accuracy (Figure 6d).

**Weight distributions**. To investigate this, we examine the distribution of parameter values in generated and training models in Figure 7. The generated weight values tend to concentrate around the average of the training values. Averaging the weights of multiple models fine-tuned from the same base model is known to lead to improved accuracy (Wortsman et al., 2022). Thus, p-diff may achieve higher accuracy in its generated models by producing interpolation of its training checkpoints.

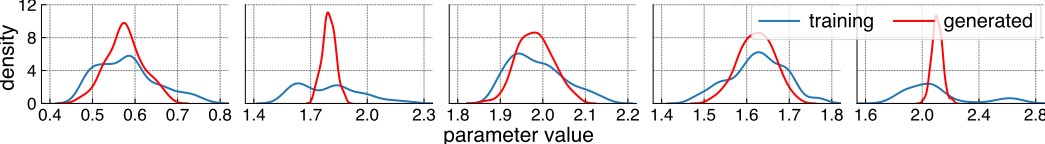

Figure 7: **Distributions of five random parameters** from the weight matrix of the first layer in the training and generated checkpoints of p-diff. The generated weights are centered around the mean of the training weights, suggesting they may be interpolations. More details are in Appendix E.2.

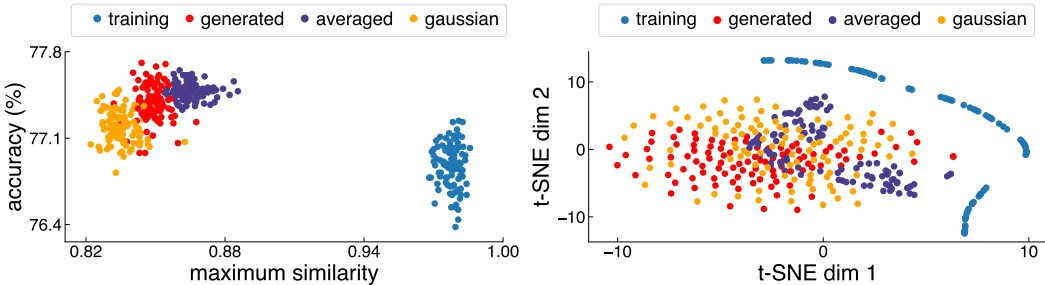

Figure 8: **P-diff generates weights with behaviors (left) and values (right) similar to interpolations of training weights**. We compare to two baselines: averaging training weights ("averaged") and sampling from a Gaussian fitted to training weights ("gaussian"). Model behavior is evaluated via accuracy and max similarity; t-SNE (Van der Maaten & Hinton, 2008) visualizes weight values.

**Interpolation baselines**. To explore this hypothesis, we generate new models using two approaches that approximate the interpolations of the training checkpoints: (1) averaging the weights of 16 randomly selected training checkpoints ("average") and (2) fitting a Gaussian distribution to the training weight values in each parameter dimension and sampling from these distributions ("gaussian").

**Behaviors and weights**. The left subplot of Figure 8 shows that the accuracy and maximum similarity of the interpolation models closely match those of p-diff. The right subplot visualizes weight distributions using t-SNE (Van der Maaten & Hinton, 2008). The weights generated by p-diff are close to weights from the above baselines, further suggesting that p-diff may primarily interpolate between training checkpoints. This interpolation occurs within a very narrow range (Appendix E).

***Summary of Section 3***. The generative models of weights produce checkpoints that closely resemble training checkpoints in both weight space and model behavior, suggesting memorization and limited novelty. Moreover, they fail to outperform simple baselines in producing new models with lower similarity to training models in model behaviors while maintaining model performance.

## 4 UNDERSTANDING MEMORIZATION IN WEIGHT GENERATION

In Section 3, we have demonstrated that conventional weight generation methods primarily memorize the training weights. In this section, we examine how modeling factors influence memorization and analyze how effectively current methods leverage weight space symmetries as structural priors.

### 4.1 LIMITED DATA AND OVERPARAMETERIZED MODELS

In image diffusion models, larger models trained on smaller datasets are more prone to memorization (Somepalli et al., 2023a; Kadkhodaie et al., 2024; Gu et al., 2025). Here, we show that memorization in weight generation can be mitigated by scaling training data, and that the overparameterized nature of these models facilitates their memorization.

**Scaling data**. We showcase on G.pt that scaling up data is a potential solution for reducing memorization in weight generation. The original G.pt model is trained on 2.1M checkpoints collected from 10228 runs. We expand the dataset to 20.4M checkpoints from 101979 runs, train a new G.pt model, and sample from it. To evaluate novelty, we measure the distances from training and generated checkpoints to their nearest training checkpoints, and show the distributions in Figure 9.

The increase in training data effectively reduces memorization without degrading the performance of the generated weights. In the original setting, the mean $L_2$ distance between generated and nearest training checkpoints (2.25) was much smaller than the mean distance between training and nearest training checkpoints (3.64). After scaling, they become nearly equal (2.78 *vs*. 2.70). Meanwhile, the mean accuracy of generated checkpoints remains unchanged (94.0% to 94.1%). However, scaling data may not be a universal solution, as Hyper-Representations does not benefit (Appendix F.1).

**Model capacity**. Generative models of weights are overparameterized; *e.g.*, despite its small training set size of 2749, HyperDiffusion has 1.4B parameters. Larger models have greater expressive

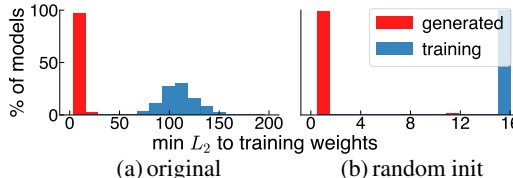

(a) original      (b) 10× data      (a) original      (b) random init

Figure 9: **Scaling data reduces memorization**. When trained on a dataset 10× the original size, G.pt produces weights with similar novelty and accuracy as an independently trained checkpoint.

Figure 10: **High model capacity enables memorization**. Even when the training weights are random initializations, HyperDiffusion produces near-identical replication of training weights.

power (Lu et al., 2017; Raghu et al., 2017), and can even fit arbitrary data without learning meaningful patterns (Zhang et al., 2017). This high capacity may be a reason why memorization occurs.

To test this, we reinitialize one or all layers of the MLP checkpoints used to train HyperDiffusion with Kaiming Uniform Initialization (He et al., 2015), and retrain HyperDiffusion. As shown in Figure 10 and Table 1, memorization is clear and consistent across all cases: generated checkpoints remain much closer to training checkpoints than training checkpoints are to one another. This indicates HyperDiffusion may not have learned the semantics of the MLPs it is trained on, but instead memorizes weight values even when those weights are random (further results in Appendix F.2).

| layers reinitialized in trained MLPs | none | 1st | 2nd | 3rd | 4th | all |
|---|---|---|---|---|---|---|
| mean dist b/w train & nearest train | 109.5 | 114.0 | 90.3 | 106.0 | 126.4 | 16.0 |
| mean dist b/w gen & nearest train | 7.0 | 9.0 | 1.5 | 2.6 | 3.6 | 0.8 |

Table 1: **HyperDiffusion replicates training weights regardless of their semantics**: even when one or all layers of the training MLPs are reinitialized with random weights.

While high model capacity is likely a contributing factor, merely reducing model size or altering other training configurations (*e.g.*, training length or regularization) does not lead to generalization and often only degrades the performance of generated models (detailed results are in Appendix F.3).

## 4.2 UNDERUSED STRUCTURAL PRIORS IN WEIGHT DATA

Computer vision researchers develop algorithms and architectures to exploit spatial, color, and texture properties of images (LeCun et al., 1989; Chen et al., 2020; Cubuk et al., 2020). Likewise, for weight data, weight space symmetries, such as permutation and scaling symmetries (introduced in Section 2.2), are promising structural priors for discriminative (Kalogeropoulos et al., 2024; Kofinas et al., 2024; Lim et al., 2024) and generative (Peebles et al., 2022; Schürholt et al., 2022) tasks.

Among the four methods we study, only G.pt and Hyper-Representations incorporate permutation symmetry. G.pt applies permutation solely as a data augmentation technique, while Hyper-Representations uses it to construct positive pairs for contrastive learning. Here, we evaluate whether these symmetry-based augmentations provide meaningful benefits for generative modeling.

**Transformation invariance**. We assess whether Hyper-Representations, trained with permutation augmentation, effectively capture weight space symmetries. Concretely, we apply function-preserving transformations to trained networks by (1) permuting neurons in hidden layers or (2) flipping weight matrix signs before and after `tanh` activations. We then reconstruct the original and transformed weights using Hyper-Representations' autoencoder, and measure the behavior similarity and accuracy difference between their reconstructions. If the autoencoder were transformation-invariant, each pair of reconstructed models would have identical accuracy and a similarity of 1.

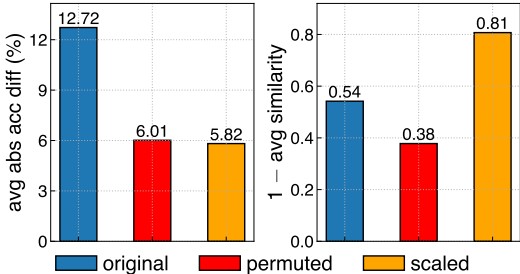

Figure 11: **Function-preserving transformations** applied to inputs of Hyper-Representations' weight autoencoder. These transformations yield reconstructed checkpoints with varied accuracy and low similarity to the reconstructions of the original inputs.

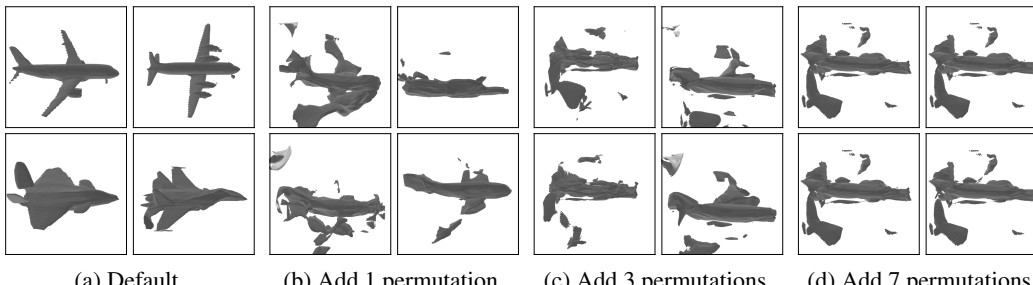

|(a) Default|(b) Add 1 permutation|(c) Add 3 permutations|(d) Add 7 permutations|

Figure 12: **HyperDiffusion fails to generate meaningful shapes when any permutation is applied**. This shows that symmetry-based augmentation is insufficient for HyperDiffusion to correctly learn the weight distributions and symmetries. Each subplot shows four random samples from a HyperDiffusion model trained with different numbers of permutations applied to the training weights.

However, as shown in the left subplot of Figure 11, reconstructions of permuted models differ in accuracy by 6.01%, and reconstructions of sign-flipped models differ by 5.82%, compared to reconstructions of their original models. For reference, the average accuracy difference between the original models is 12.72%. Similarly, the right subplot shows that the error similarity between reconstructions of original and permuted models is 0.62, and for sign-flipped models, the similarity is only 0.19. These results indicate that the autoencoder fails to fully capture weight space symmetries, despite being trained with a contrastive loss that enforces permutation invariance.

**Permutation augmentation**. We investigate whether applying data augmentation based on weight symmetries can reduce the memorization in HyperDiffusion. Specifically, we add 1, 3, and 7 random weight permutations during training, effectively enlarging the dataset by factors of $\times 2$, $\times 4$, and $\times 8$, respectively. To ensure convergence, we double the training epochs.

Figure 12 shows the shapes generated by the resulting models. Even when we only add a single permutation, HyperDiffusion fails to produce meaningful shapes. As the number of added permutations increases (*e.g.*, to three), HyperDiffusion fails to converge during training (see Appendix G).

**Discussion**. These findings suggest that applying weight space symmetries merely as data augmentations is insufficient for encoding such symmetries into generative models and may even make the training distribution harder to model. Future generative modeling methods may benefit from architectures that are invariant to weight symmetries by design. For example, prior work on classifying or editing network weights has shown that neural networks can be represented as graphs and processed by graph neural networks explicitly designed to respect weight symmetries (Kalogeropoulos et al., 2024; Kofinas et al., 2024; Lim et al., 2024).

***Summary of Section 4***. Limited data and overparameterized models are two potential causes for memorization in weight generation. Further, existing data augmentation methods based on weight symmetries are insufficient for models to learn the weight distributions and symmetries. Thus, explicitly integrating the structural priors of weight data into the model design may be beneficial.

## 5 CONCLUSION

We provide evidence that current generative modeling methods for weights primarily memorize training data rather than generating truly novel network weights. Our analysis shows that generated checkpoints are close replicas or interpolations of training checkpoints, with similar weight values and model behaviors. We find that the generation methods offer no clear advantage over simple baselines to create novel, high-performing models. Factors such as limited data, overparameterized models, and the underuse of structural priors in weight data likely contribute to this memorization.

Our findings emphasize the need for careful model design and evaluation of memorization in generative modeling, particularly as these models expand to new modalities and tasks. We hope this work can inspire future research to address the memorization issues, and further explore the practical applications of generative models for weight data and beyond.

REPRODUCIBILITY STATEMENT

Our analysis and visualization code, along with detailed instructions, can be accessed through this anonymous link. All of our results are reproducible on a single NVIDIA A100 GPU.

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

APPENDIX

# A    EXPERIMENTAL SETUP FOR EACH METHOD

In this paper, we analyze four representative generative modeling methods for neural network weights, under their primary experimental setups. Here, we expand on Section 2.1 to provide more details about the training, inference, and analysis of the four methods.

**Hyper-Representations** provides, for each of MNIST, SVHN, CIFAR-10, and STL-10, a dataset of thousands of classification model checkpoints, together with a pre-trained generative model trained on that checkpoint dataset. We evaluate the pre-trained Hyper-Representations autoencoder checkpoint for SVHN classification model weights. Unless explicitly stated otherwise, the training weights referenced throughout the paper refer to the reconstructions of the original training weights produced by the Hyper-Representations autoencoder.

Hyper-Representations save one checkpoint at each of epochs 21 to 25 in every classification model training run when creating its datasets of checkpoints. However, when calculating the $L_2$ distance from training and generated checkpoints to their nearest training checkpoints, we only use training checkpoints from the 25th epoch. This is to ensure that the distances between training checkpoints correctly represent distances between models independently trained from scratch.

**G.pt** provides datasets of 2.1M to 11.3M checkpoints and pre-trained generative models of weights for MNIST and CIFAR-10 classification models, as well as Cartpole reinforcement learning models. We evaluate the pre-trained generative model for MNIST classification model weights.

Although G.pt's training procedure uses 200 checkpoints from each MNIST training run, to reduce computational cost, we use only the final checkpoint from each run as the training weights throughout our paper. This would not underestimate memorization, because the one-step generation of G.pt explicitly prompts the generative model to produce MNIST classification model weights with zero test loss, making the generated weights more similar to final checkpoints than to earlier ones.

**HyperDiffusion** trains an unconditional diffusion model on MLPs that each represent a neural occupancy field of a unique 3D shape. These MLPs are trained to map 3D coordinates to occupancy values. Meshes can be extracted from the MLPs using Marching Cubes (Lorensen & Cline, 1987).

HyperDiffusion is applied to MLPs trained on the car, chair, and airplane categories of the ShapeNet dataset (Chang et al., 2015), as well as to MLPs representing 4D animation sequences. However, the dataset of MLP checkpoints and the corresponding generative model of MLP weights were only released for airplane shapes. Thus, our analysis is focused only on this experimental setting.

**P-diff** is applied to multiple image classification datasets and model architectures. However, the analysis in the original paper focuses on the setting of generating the last two batch normalization layers for a CIFAR-100 classification model of ResNet-18 architecture. Accordingly, we evaluate p-diff under this setting. Since no weight datasets or pre-trained generative models are released, we follow the official codebase to collect the training weights and train the generative model ourselves.

## B   RESULTS ON NON-PARAMETRIC TEST FOR DATA-COPYING DETECTION

Meehan et al. (2020) proposed a non-parametric test for detecting memorization in generative models. The test requires the training set, a holdout set $P_n$ of size $n$, and a set of generated samples $Q_m$ of size $m$. The $n + m$ samples in $P_n \cup Q_m$ are sorted by their distance to the nearest training data point, with rank 1 assigned to the closest sample and rank $n + m$ to the farthest. Let $R_{Q_m}$ denote the sum of the ranks of all generated samples $Q_m$. The $U$ statistic (Mann & Whitney, 1947) is

$$U_{Q_m} = R_{Q_m} - \frac{m(m+1)}{2},\tag{1}$$

which is then normalized to obtain the $z$-scored statistic $Z_U$:

$$Z_U = \frac{U_{Q_m} - \mu_U}{\sigma_U},\tag{2}$$

where $\mu_U = \frac{mn}{2}$ and $\sigma_U = \sqrt{\frac{mn(m+n+1)}{12}}$. Intuitively, $Z_U \ll 0$ indicates memorization (data-copying), while $Z_U \gg 0$ suggests that the generative model underfits the training data.

We apply the $Z_U$ metric to the original generative models of weights from the four studied methods, as well as to the models trained in our ablation experiments. This provides additional quantitative evidence regarding whether the models memorize or generalize. Since each method provides a different number of holdout samples, the $Z_U$ values are not directly comparable across methods.

### B.1   DATA COPYING TEST FOR ORIGINAL MODELS

In Section 3.1, our visualization showed that, except for p-diff, whose training checkpoints are saved consecutively in the same run and are of very low diversity, the generated checkpoints of all other methods are much closer to the training checkpoints than the training checkpoints are to one another.

Here, we measure the $Z_U$ score based on the same $L_2$ distance metric between checkpoints for Hyper-Representations, G.pt, and HyperDiffusion, and find that the $Z_U$ (-13.6, -8.5, and -30.8, respectively) scores are significantly smaller than 0, indicating severe memorization. This confirms the results in Section 3.1.

### B.2   DATA COPYING TEST FOR MODELS FROM ABLATION EXPERIMENTS

In Section 4.1, we identified limited dataset size and model overparameterization as potential reasons why memorization occurs. Specifically, we showed that scaling up the training data reduces memorization in G.pt, while HyperDiffusion is capable of memorizing random weights. Here, we further confirm these trends using the $Z_U$ score.

**Scaling data for G.pt**. We measure the $Z_U$ score before and after increasing the training dataset size of G.pt from 2.1M to 20.4M samples. The $Z_U$ score rises from -8.5 to 3.5, indicating that scaling data effectively mitigates memorization in G.pt.

| training dataset size | 2.1M | 20.4M |
|---|---|---|
| mean dist b/w train & nearest train | 3.64 | 2.70 |
| mean dist b/w gen & nearest train | 2.25 | 2.78 |
| $Z_U$ score (Meehan et al., 2020) | -8.5 | 3.5 |

Table 2: **Scaling up training data can reduce memorization in G.pt**. The increase to positive $Z_U$ score after scaling up data confirms the reduced memorization observed in Figure 9.

**Training HyperDiffusion on random weights**. In Section 4.1, we trained HyperDiffusion on the original dataset of checkpoints with one or all layers randomly reinitialized. We report the $Z_U$ score of the original and newly-trained HyperDiffusion models in Table 3.

We find that all models have a $Z_U$ score of -30.8, the lowest possible score when $n = 606$ and $m = 666$. This is because, in every case, each generated checkpoint is closer to a training checkpoint than any training checkpoint is to another training checkpoint. This further confirms that HyperDiffusion closely memorizes the training checkpoints regardless of the semantics of the weights.

| layers reinitialized in trained MLPs | none | 1st | 2nd | 3rd | 4th | all |
|---|---|---|---|---|---|---|
| mean dist b/w train & nearest train | 109.5 | 114.0 | 90.3 | 106.0 | 126.4 | 16.0 |
| mean dist b/w gen & nearest train | 7.0 | 9.0 | 1.5 | 2.6 | 3.6 | 0.8 |
| $Z_U$ score (Meehan et al., 2020) | -30.8 | -30.8 | -30.8 | -30.8 | -30.8 | -30.8 |

Table 3: **HyperDiffusion memorizes weights regardless of their semantics.** The consistently low $Z_U$ score confirms that severe memorization occurs in all cases, as observed in Table 1.

# C  ADDITIONAL VISUALIZATION OF MODEL WEIGHTS AND BEHAVIORS

## C.1  ADDITIONAL RANDOM EXAMPLES OF WEIGHT HEATMAPS

In Section 3.1, we showed the values of random parameters in generated checkpoints and their nearest training checkpoints for all four methods, to demonstrate the memorization in weight space. Due to space constraints, we presented only three random examples per method. In Figure 13, we provide heatmap visualizations of eight additional random generated checkpoints and their nearest training checkpoints for each method. Consistently, we observe that for almost every generated checkpoint, there exists at least one training checkpoint with highly similar weights.

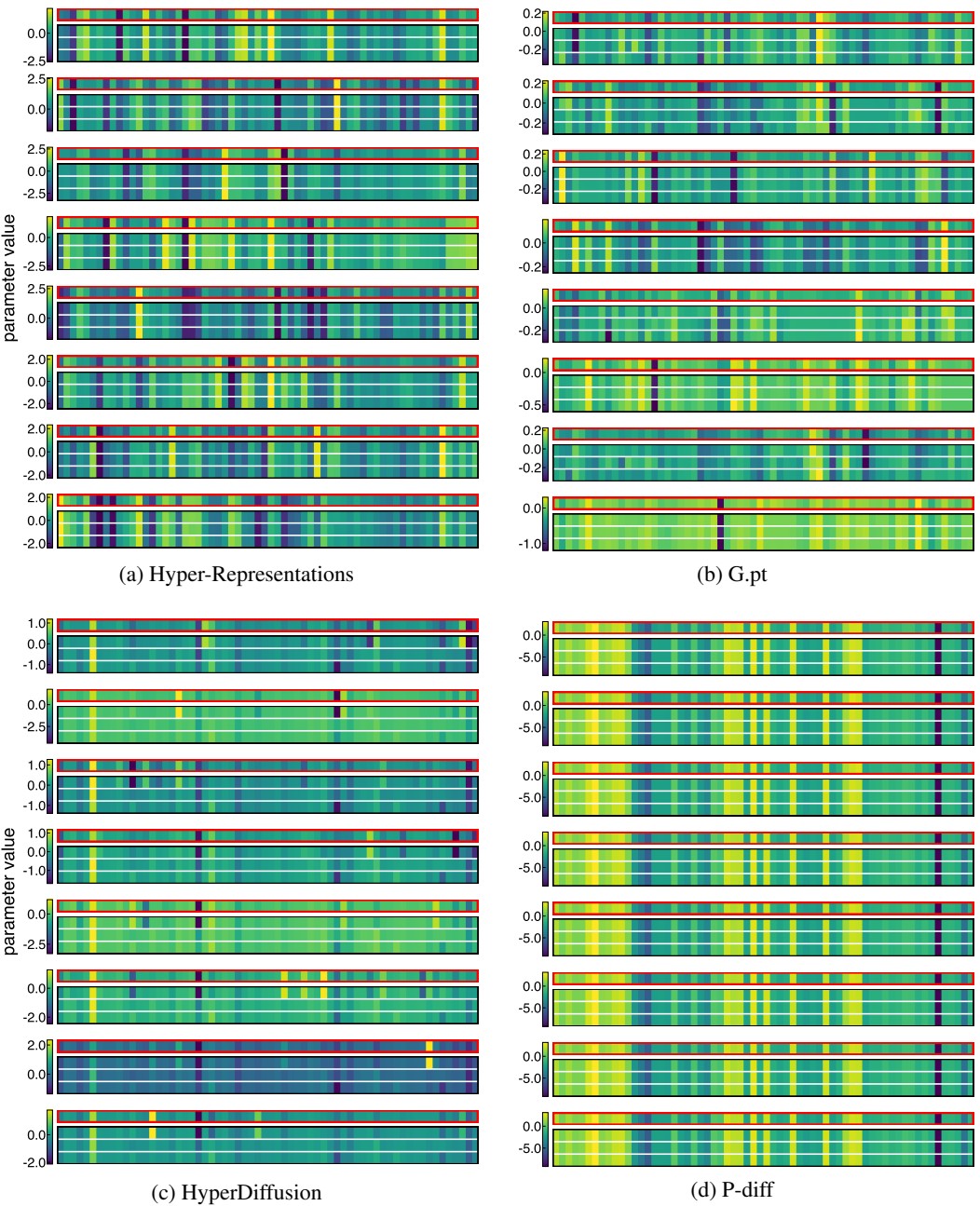

(a) Hyper-Representations

(b) G.pt

(c) HyperDiffusion

(d) P-diff

Figure 13: **Additional random examples of weight heatmap** for generated models and their nearest training models.

## C.2 Heatmaps by Percentile of Distance to Nearest Training Checkpoint

In Section 3.1 and Appendix C.1, we showed weight heatmaps of random generated checkpoints and their nearest training checkpoints. Here, we further rank the generated checkpoints by their $L_2$ distance to the nearest training checkpoint and present weight heatmaps at different percentiles in Figure 14. A lower percentile corresponds to a smaller distance to the nearest training checkpoint.

Consistent with our earlier findings, the generated weights from Hyper-Representations are nearly identical to their nearest training weights across all percentiles. Similarly, for G.pt and HyperDiffusion, all generated checkpoints are highly similar to their nearest training checkpoints, except at the 100th percentile, which show moderate differences. For p-diff, across all percentiles, all training and generated checkpoints are nearly identical. As noted in Section 3, this is likely because its training checkpoints are saved from consecutive steps within the same training run, resulting in low diversity.

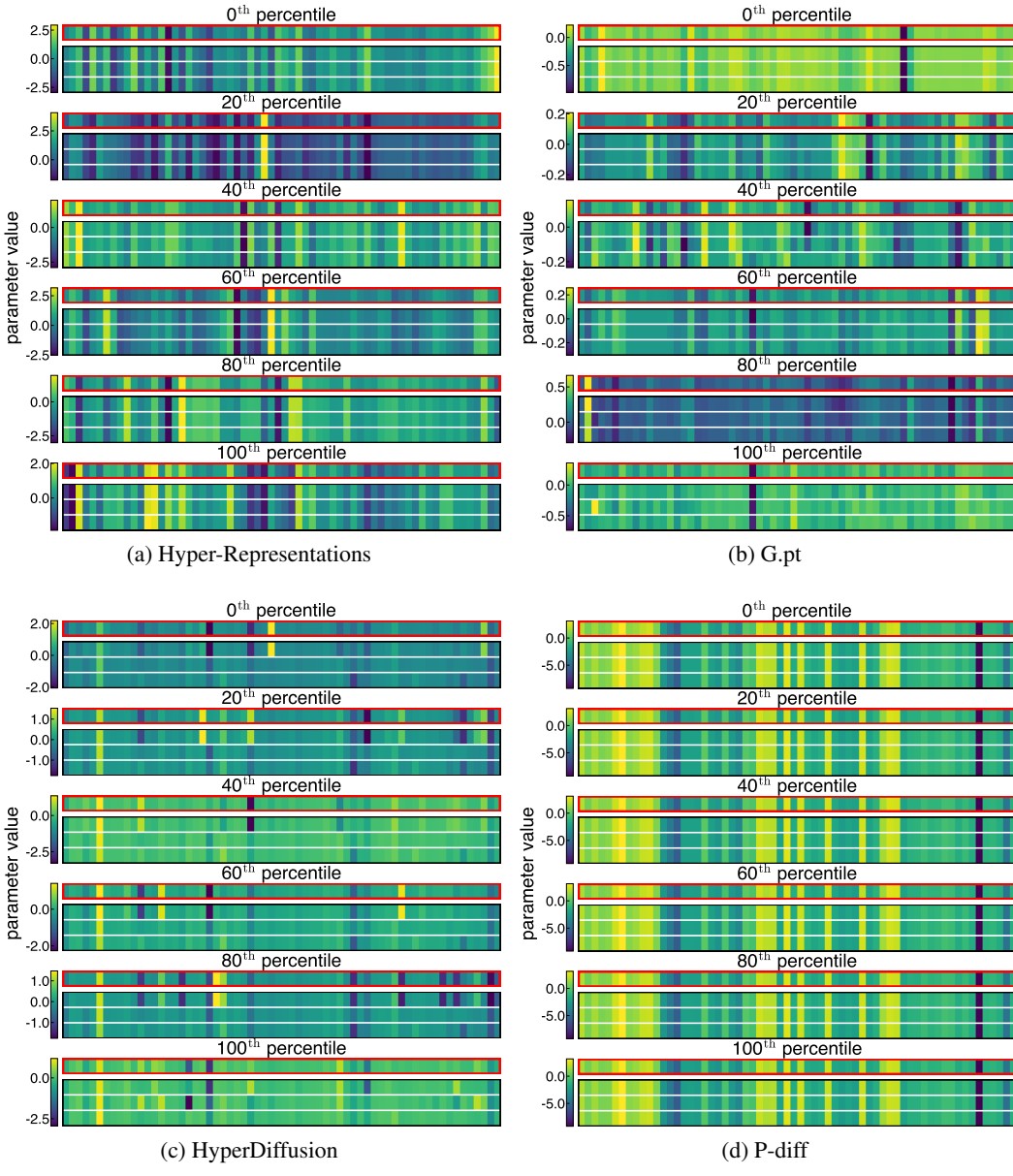

Figure 14: **Heatmaps of generated checkpoints at different percentiles of distance to the nearest training checkpoint**. Results are consistent with those observed for random generated weights: all generated checkpoints closely resemble their nearest training checkpoints, except for those at the 100th percentile in G.pt and HyperDiffusion, which show moderate differences.

## C.3 Additional Random Examples of Model Outputs

In Section 3.2, we visualized the decision boundaries or reconstructed 3D shapes of generated models and their nearest training models, to demonstrate their high similarity in model behaviors. Due to space constraints, we presented only three random examples per method. In Figure 15, we provide visualizations of model outputs for nine additional random generated checkpoints per method and the nearest training checkpoint to each of them. These results further confirm that the generated checkpoints closely resemble their nearest training checkpoints not only in weight space but also in model behavior.

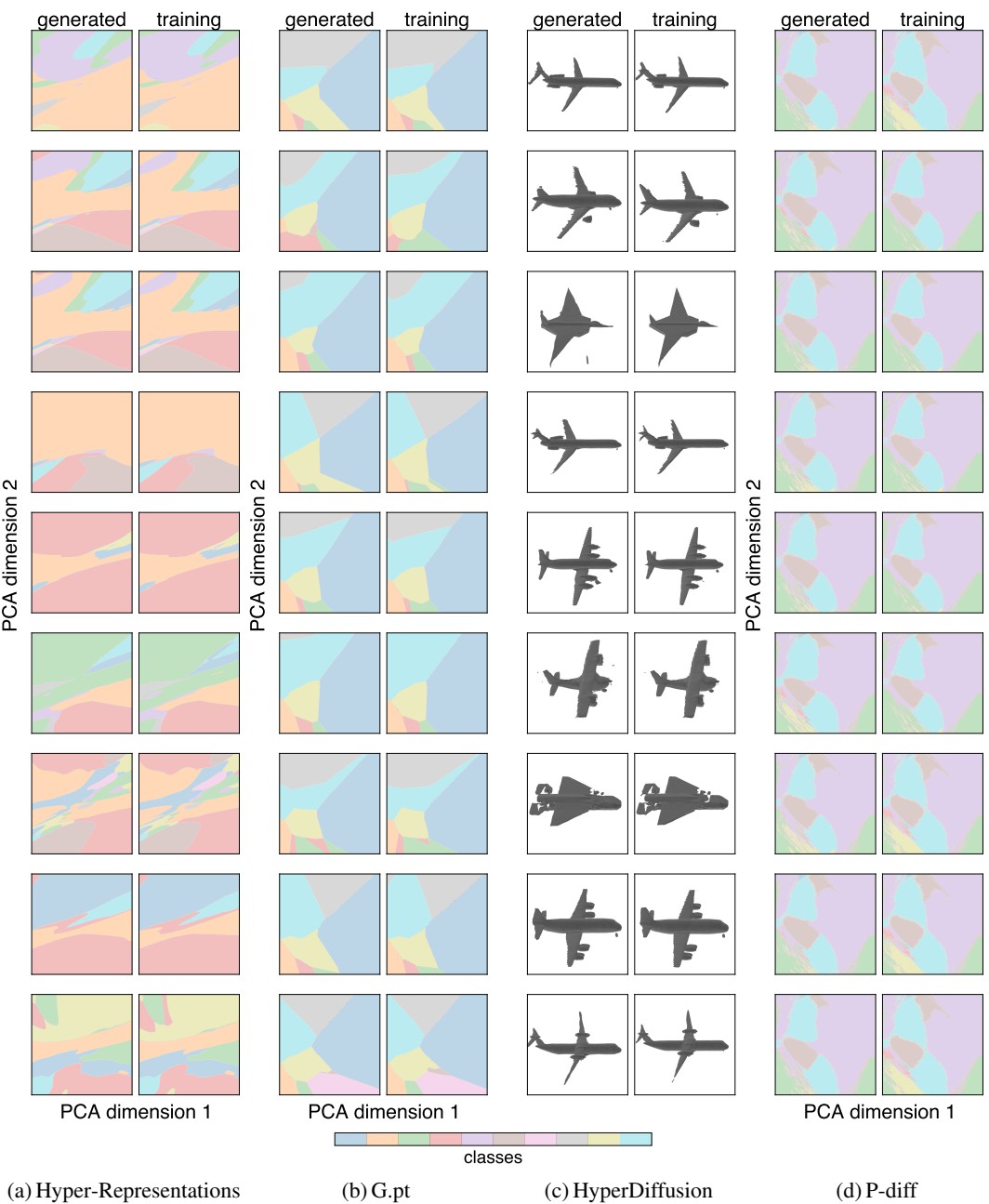

(a) Hyper-Representations  (b) G.pt  (c) HyperDiffusion  (d) P-diff

Figure 15: **Addition random examples of decision boundaries and reconstructed meshes** of generated models and their nearest training models.

### C.4 MODEL OUTPUTS BY PERCENTILE OF DISTANCE TO NEAREST TRAINING CHECKPOINT

In Section 3.2 and Appendix C.3, we showed model outputs from random generated checkpoints and their nearest training checkpoints. Here, we further rank the generated checkpoints by their $L_2$ distance to the nearest training checkpoint and present model outputs at different percentiles in Figure 16. A lower percentile corresponds to a smaller distance to the nearest training checkpoint.

Consistent with our earlier findings, the behaviors of generated models from Hyper-Representations are nearly identical to their nearest training weights across all percentiles. Similarly, for G.pt and HyperDiffusion, all generated checkpoints produce outputs highly similar to their nearest training checkpoints, except those at the 100th percentile, which show moderate differences. We note that the HyperDiffusion-generated checkpoint at the 100th percentile is of low quality (as seen in the degraded shape it reconstructs to) and thus cannot be matched to any training checkpoint. For p-diff, across all percentiles, all training and generated checkpoints' outputs are highly similar.

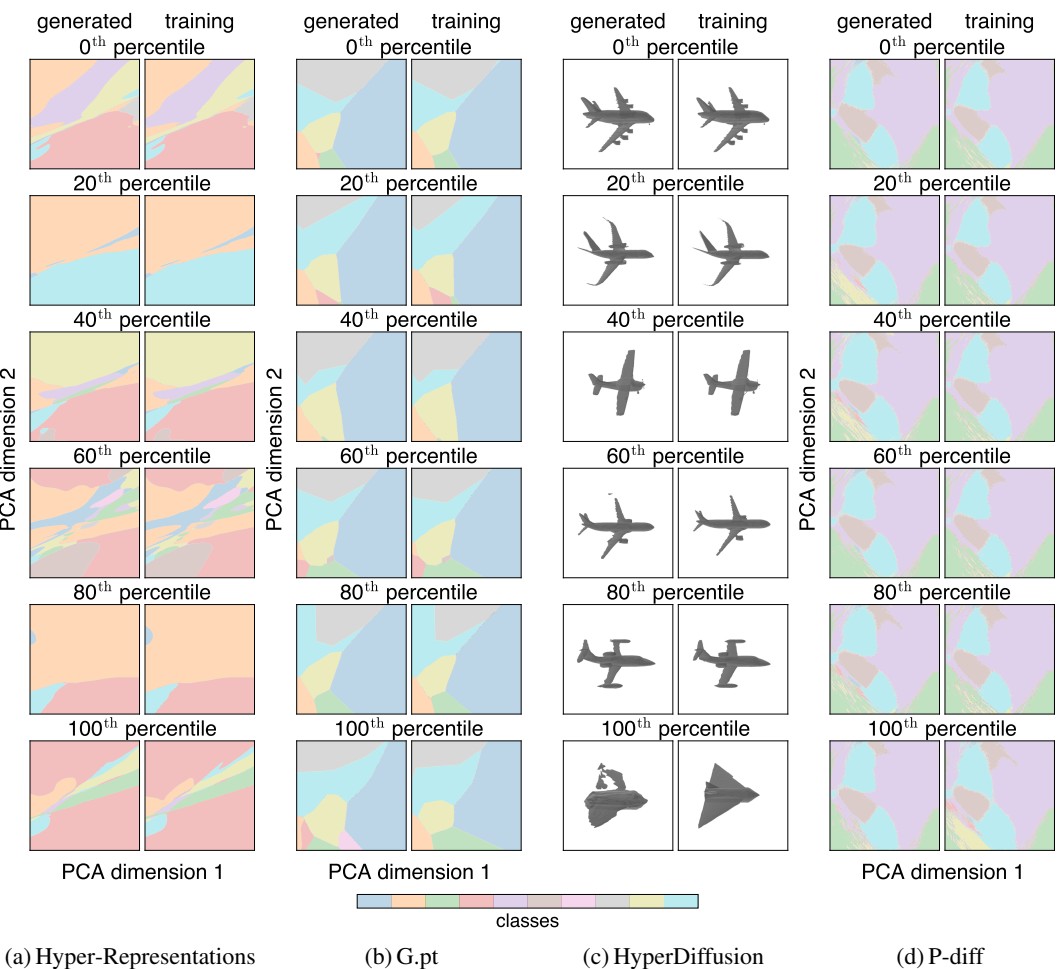

(a) Hyper-Representations      (b) G.pt      (c) HyperDiffusion      (d) P-diff

Figure 16: **Decision boundaries and reconstructed meshes of generated checkpoints at different percentiles of distance to the nearest training checkpoint.** Results are consistent with those observed for random generated weights: all generated checkpoints closely resemble their nearest training checkpoints in model outputs, except at the 100th percentile in G.pt and HyperDiffusion, where the lower quality of the generated checkpoints may account for the observed differences.

# D  ADDITIONAL INFORMATION ON MEMORIZATION IN MODEL BEHAVIORS

## D.1  METRIC FOR CHECKPOINT SIMILARITY BASED ON INCORRECT PREDICTIONS

In Section 3.2, we used the metric from Wang et al. (2024) to quantify the similarity between two classification model checkpoints. The metric measures similarity based on the Intersection over Union (IoU) of the sets of incorrect predictions made by the two model checkpoints. Formally, it is defined as follows:

$$
\begin{aligned}
I_1 &= \{ i \in \{1, \ldots, N\} \mid M_1(X_i) \neq y_i \}, \\
I_2 &= \{ i \in \{1, \ldots, N\} \mid M_2(X_i) \neq y_i \}, \\
\text{IoU}(M_1, M_2) &= \frac{|I_1 \cap I_2|}{|I_1 \cup I_2|},
\end{aligned}
\tag{3}
$$

where $\{(X_i, y_i)\}_{i=1}^N$ represents the test set on which the model checkpoints are evaluated. The sets $I_1$ and $I_2$ contain the indices of test samples for which model checkpoints $M_1$ and $M_2$ make incorrect predictions, respectively.

## D.2  ADDITIONAL INFORMATION ON THE NOISE-ADDITION BASELINE

In Section 3.2, we introduced a noise-addition baseline to compare with the generated models in terms of performance and novelty. For Hyper-Representations, whose KDE sampling method is based on the top 30% highest-accuracy training checkpoints, we apply noise to reconstructions of a random subset of these highest-accuracy checkpoints to ensure a fair comparison. For all other methods, we apply noise to checkpoints uniformly sampled from all training checkpoints.

## D.3  ALTERNATIVE SIMILARITY METRIC: OVERLAP IN CLASSIFICATION PREDICTIONS

For classification models, the percentage of test set predictions they agree on provides an intuitive measure of their similarity in behavior. Table 4 shows the prediction overlap between classification model weights generated by Hyper-Representations, G.pt, and P-diff and their nearest training checkpoints under $L_2$ distance, along with prediction overlap between training models and their nearest neighbors (excluding self-comparisons) for comparison. As in Section 3, for Hyper-Representations, we use reconstructed training weights rather than the original ones.

| method | Hyper-Representations | G.pt | P-diff |
|---|---|---|---|
| mean accuracy of training models | 51.3 | 94.5 | 76.9 |
| pred overlap b/w training & nearest training | 75.6 | 97.9 | 91.4 |
| pred overlap b/w generated & nearest training | 98.5 | 98.2 | 93.5 |

Table 4: **Classification predictions highly overlap between generated and training models**. This shows that the generated models highly resemble the behaviors of training models.

Across all methods, generated models show higher prediction overlap with their nearest training models than training models do. This high overlap suggests that the generated models closely resemble the training models in behavior. However, we note that prediction overlap can be strongly influenced by accuracy: two models with accuracy $x$ will have a minimum overlap of $\max(2x-1, 0)$.

# E   ADDITIONAL INFORMATION ON THE NOVELTY OF P-DIFF'S GENERATED MODELS

All training and generated checkpoints of p-diff exhibit highly similar weight values (Figures 2 and 13) and decision boundaries (Figures 5 and 15). Yet, unlike other methods, p-diff achieves a better accuracy-novelty trade-off than noise addition (Figure 6), and its generated models are often farther from the nearest training model than training models are from one another (Figure 4).

This may be explained by p-diff's training checkpoints being saved from consecutive steps in the same training run, which results in significantly lower diversity in training models, compared to other methods that sample checkpoints across different runs. Consequently, p-diff may be interpolating within a narrow region of the weight space, which still *appears* novel relative to its low-diversity training distribution.

To investigate this, we analyze the weight distribution of p-diff's training and generated checkpoints, in comparison with models trained from scratch using different random seeds.

## E.1   COMPARISON WITH MODELS TRAINED FROM SCRATCH

We train 20 models from scratch using different random seeds, with the same architecture (ResNet-18) and downstream task (CIFAR-100) as p-diff. The training recipe, shown in Table 5, is tuned so that the final accuracies of these models (75.4% ± 0.3%) approximately match those of p-diff's training checkpoints (76.8% ± 0.2%).

| config | value |
| --- | --- |
| optimizer | AdamW (Loshchilov & Hutter, 2019) |
| learning rate | 5e-4 |
| weight decay | 5e-4 |
| optimizer momentum | $\beta_1, \beta_2 = 0.9, 0.999$ |
| batch size | 128 |
| learning rate schedule | cosine decay |
| training epochs | 300 |
| augmentation | `RandomResizedCrop` (Szegedy et al., 2015) & `RandAug` (9, 0.5) (Cubuk et al., 2020) |

Table 5: **Training recipe** for CIFAR-100 classification models trained from scratch.

In Figure 17, we visualize the weights of 20 randomly selected checkpoints from each group: p-diff training checkpoints, p-diff generated checkpoints, and models trained from scratch. We sample the same 64 parameter indices across all models. Both the training and generated checkpoints from p-diff exhibit minimal variation, while the from-scratch models display substantially greater diversity in parameter values.

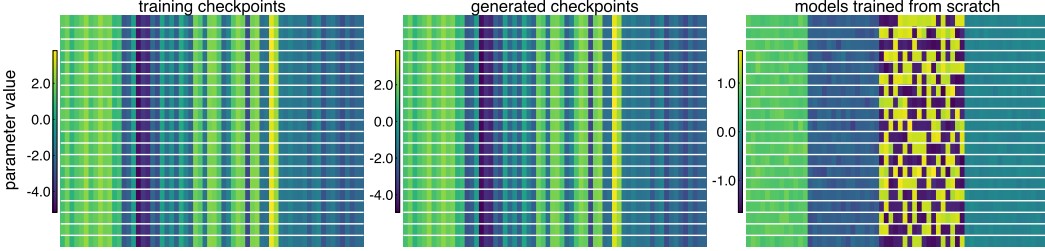

Figure 17: **P-diff's training and generated checkpoints show limited diversity compared to models trained from scratch.**. Each row (separated by white lines) is a model checkpoint; each column is a randomly selected parameter index. The same indices are used across all three subplots.

We further quantify the weight space diversity of p-diff's training and generated checkpoints, compared to models trained from scratch. As shown in Table 6, both training and generated checkpoints

of p-diff occupy a narrow region of the weight space, with low pairwise and nearest-neighbor distances. In contrast, models trained independently from scratch exhibit much higher weight variation.

| case | mean $L_2$ distance |
|---|---|
| b/w all pairs of training checkpoints | 6.9 |
| b/w all pairs of training and generated checkpoints | 6.3 |
| b/w all pairs of from-scratch models | 46.1 |
| from training checkpoints to nearest training checkpoints | 0.3 |
| from generated checkpoints to nearest training checkpoints | 5.4 |
| from from-scratch models to nearest from-scratch models | 44.1 |

Table 6: **Distances among p-diff's training and generated checkpoints are much smaller than the distances among from-scratch models**. This shows that p-diff's training and generated checkpoints occupy a narrow range in weight space compared to models trained from scratch.

These results confirm that p-diff's training and generated checkpoints occupy a highly constrained region in weight space, substantially narrower than the region spanned by independently trained models. Thus, although p-diff appears to interpolate between training checkpoints (Figures 7 and 8), the training checkpoints themselves lack diversity. As a result, the interpolation occurs within a narrow subspace and does not reflect meaningful generalization beyond the training data.

### E.2 ADDITIONAL INFORMATION ON P-DIFF WEIGHT VALUE DISTRIBUTION

In Section 3.2, we showed that the parameter values in checkpoints generated by p-diff tend to center around the average of the parameter values in training checkpoints, using smoothed weight distribution curves. Figure 18 presents the same plot without smoothing, confirming that the moderate smoothing does not affect the observed trends.

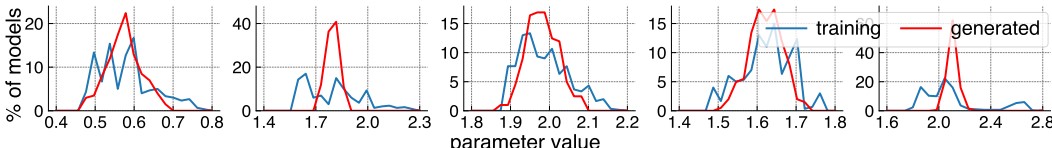

Figure 18: **Distributions of 5 randomly selected parameters** from the weight matrix of the first layer in the training and generated checkpoints of p-diff. This figure corresponds to Figure 7 but without smoothing applied to the distribution curves.

We further extend this analysis by visualizing the parameter value distributions of 50 randomly selected entries from the first-layer weight matrix in both training and generated checkpoints. As shown in Figure 19, the concentration of generated weights around the mean of training weights persists across this larger set of parameters.

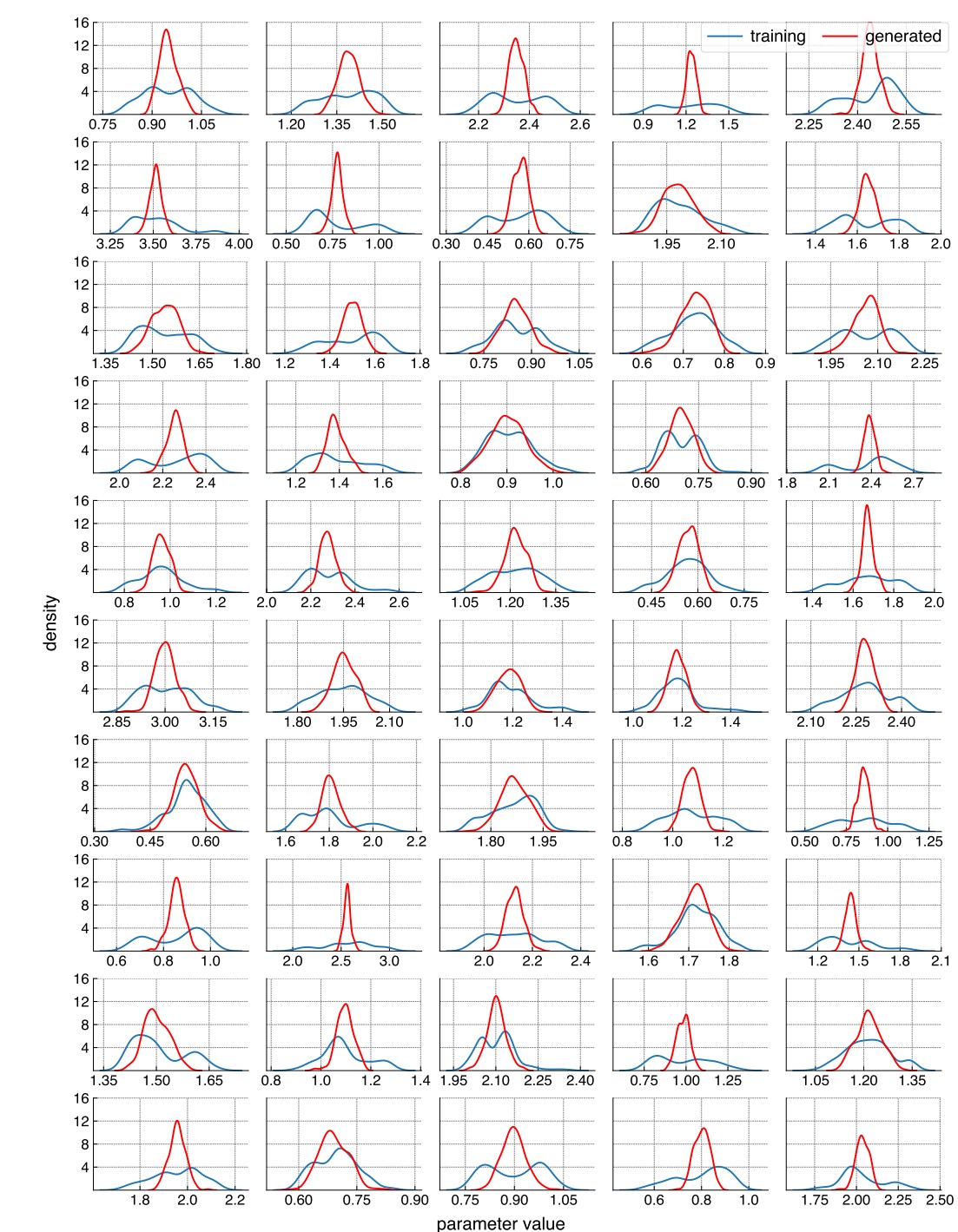

Figure 19: **Distributions of 50 randomly selected parameters** from the weight matrix of the first layer in the training and generated checkpoints of p-diff. This figure extends the analysis of Figure 7 to a broader set of parameters, further confirming the observed trend of generated weight values concentrating around the average of the training values.

# F ADDITIONAL INFORMATION ON THE IMPACT OF TRAINING CONFIGURATIONS ON MEMORIZATION

In Section 4.1, we demonstrated that limited data and overparameterized models likely contribute to memorization. Here, we extend this analysis by testing additional settings and modeling factors beyond data and model size. The $Z_U$ score used in this section is introduced in Appendix B.

## F.1 DATA SCALING

In Section 4.1, we showed that scaling up training data for G.pt can effectively mitigate memorization. Here, we further explore scaling data for Hyper-Representations.

| training dataset size | 2896 | 24136 |
|---|---|---|
| mean dist b/w train & nearest train | 49.97 | 48.17 |
| mean dist b/w gen & nearest train | 8.11 | 9.24 |
| $Z_U$ score (Meehan et al., 2020) | -13.6 | -13.6 |

Table 7: **Scaling up training data does not reduce memorization in Hyper-Representations**. After scaling up, the distances between training and generated checkpoints remain much smaller than the distances among training checkpoints, and $Z_U$ remains unchanged.

Concretely, we increased the number of training checkpoints from 2896 to 24136. However, as shown in Table 7, generated checkpoints remain far closer to training checkpoints than training checkpoints are to each other. The unchanged $Z_U$ score after scaling further confirms that scaling data does not mitigate memorization in Hyper-Representations.

While this result does not rule out the possibility that scaling to much larger datasets might eventually reduce memorization, it suggests that there may be more fundamental modeling issues at play (*e.g.*, the lack of designs explicitly integrating the properties of weight data, as discussed in Section 4.2), beyond insufficient data.

## F.2 MODEL CAPACITY

In Section 4.1, we showed that HyperDiffusion can fully memorize its training checkpoints, even when one or all layers are reinitialized with random weights, suggesting that it simply memorizes weights without capturing meaningful patterns. Here, we further demonstrate this by training HyperDiffusion on MLPs trained with different training lengths.

By default, HyperDiffusion trains the first MLP model from random initialization, and subsequent MLP models are initialized from the trained weights of the first model. To enable a fairer comparison across training lengths, we instead train all MLP models from scratch.

| training length | 0 epoch | 200 epochs | 400 epochs | full |
|---|---|---|---|---|
| mean dist b/w train & nearest train | 16.0 | 115.7 | 129.4 | 126.8 |
| mean dist b/w gen & nearest train | 0.8 | 2.8 | 3.7 | 5.7 |
| $Z_U$ score (Meehan et al., 2020) | -30.8 | -30.8 | -30.8 | -30.8 |

Table 8: **HyperDiffusion memorizes MLP weights regardless of their training length**. This suggests that memorization occurs independent of the semantics of the weights.

Following HyperDiffusion's codebase, a full MLP training run ends when the training loss fails to improve for 50 consecutive epochs. Across 500 independent runs, the average training length is 667.7 epochs. We collect new datasets of checkpoints by training all MLPs for 200 and for 400 epochs, and then train a HyperDiffusion on each dataset. The evaluation results are shown in Table 8.

## F.3 OTHER MODELING FACTORS

Modeling choices such as training duration, model architecture, and regularization strategy have been shown to significantly impact memorization in image diffusion models (Somepalli et al.,

| | Hyper-Representations | | | G.pt | | | HyperDiffusion | | |
|---|---|---|---|---|---|---|---|---|---|
| | #params | acc.↑ | $L_2$ ↑ | #params | acc.↑ | $L_2$ ↑ | #params | MMD↓ | $L_2$ ↑ |
| *baseline* | | | | | | | | | |
| training | – | 65.2 | 50.0 | – | 94.4 | 3.64 | – | 0.026 | 109.5 |
| default gen | 223M | 57.5 | 8.11 | 378M | 94.0 | 2.25 | 1.4B | 0.036 | 7.03 |
| *training epochs* | | | | | | | | | |
| 33.3% | 223M | 33.8 | 7.86 | 378M | 94.0 | 2.40 | 1.4B | 0.035 | 7.40 |
| 50.0% | 223M | 47.1 | 7.97 | 378M | 94.0 | 2.25 | 1.4B | 0.034 | 4.74 |
| *model size* | | | | | | | | | |
| +dim & head | 359M | 50.1 | 8.06 | 579M | 93.6 | 2.12 | 2.1B | 0.034 | 2.69 |
| +layer | 362M | 55.9 | 8.09 | 605M | 93.9 | 2.08 | 2.0B | 0.036 | 3.32 |
| −dim & head | 118M | 44.1 | 7.93 | 220M | 93.6 | 2.51 | 0.8B | 0.039 | 22.47 |
| −layer | 154M | 42.2 | 7.93 | 208M | 86.6 | 3.70 | 1.0B | 0.033 | 3.17 |
| *regularization* | | | | | | | | | |
| +10% dropout | 223M | 53.9 | 7.76 | 378M | 93.7 | 2.27 | 1.4B | 0.035 | 5.10 |
| +20% dropout | 223M | 44.7 | 7.26 | 378M | 92.5 | 3.13 | 1.4B | 0.034 | 6.08 |
| +Gaussian noise | 223M | 57.8 | 8.14 | 378M | 92.9 | 2.37 | 1.4B | 0.033 | 3.24 |

Table 9: **Modeling changes do not effectively mitigate memorization**: modifications known to reduce memorization in image diffusion fail to meaningfully improve the novelty of generated weights (measured via $L_2$ to nearest training model) without degrading performance. The resulting changes in $L_2$ for generated weights are often much smaller than the gap in $L_2$ between the two baselines.

2023b; Yoon et al., 2023; Kadkhodaie et al., 2024; Gu et al., 2025). In Section 4.1, we also showed that the large size of the generative models of weights likely contributes to memorization. Here, we investigate whether adjusting these factors suffices to mitigate the memorization in generative models of weights.

**Quantitative metrics**. In Section 3, we apply various metrics and baselines to demonstrate the memorization in weight space and model behaviors. Here, we measure the mean $L_2$ distance between generated models and their nearest training models, as a simple proxy to quantify the novelty of generated weights. However, a high $L_2$ distance to training weights may also arise from low-quality weight generations. Thus, we also evaluate model performance: accuracy for classification models and Minimum Matching Distance (under Chamfer Distance) for neural field models. These quantitative evaluations of generative models under varying modeling factors are shown in Table 9. We report the metrics for training weights and generated weights under default settings as baselines.

**Training epochs**. Reducing training epochs tends to lessen memorization in generative models (Somepalli et al., 2023b; Yoon et al., 2023; Gu et al., 2025). We shorten training to 1/2 and 1/3 of the original length. Nonetheless, this has minimal impact on the $L_2$ distances and the quality of generated weights for G.pt and HyperDiffusion, while significantly degrading the accuracy of models produced from Hyper-Representations.

**Model size**. The size of a generative model can influence its sample quality and generalization (Dhariwal & Nichol, 2021; Nichol & Dhariwal, 2021). We vary the model size by increasing or decreasing its depth (*i.e.*, number of layers) and width (*i.e.*, model dimensions). However, across the three methods, changing the model size does not meaningfully increase the $L_2$ distances without compromising the generated models' performance.

**Regularization**. Regularization techniques have long been leveraged to prevent models from overfitting to the training set (Srivastava et al., 2014; Szegedy et al., 2016; Pereyra et al., 2017). Here, we apply dropout (Srivastava et al., 2014) and inject random Gaussian noise into the training weights. Yet, these only result in minor changes to sample quality and $L_2$ distances.

**Discussion**. Modeling factor adjustments common in image diffusion cannot alleviate the memorization issue: none substantially improved the novelty of the generated weights without degrading performance. Notably, the changes in $L_2$ distance resulting from these variations were much smaller than the original gap between $L_2$ measured on training weights and on generated weights.

## G ADDITIONAL INFORMATION ON PERMUTATION AUGMENTATION FOR HYPERDIFFUSION

In Section 4.2, we investigated whether adding permutation augmentations to the training data of HyperDiffusion reduces memorization. Specifically, we added 1, 3, and 7 random weight permutations during training, effectively enlarging the dataset by factors of $\times 2$, $\times 4$, and $\times 8$, respectively.

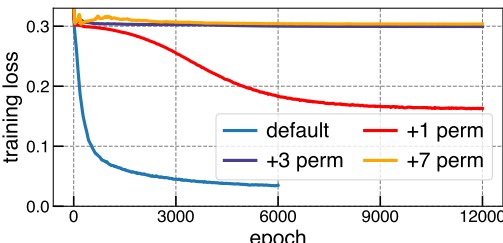

Figure 20: **HyperDiffusion fails to converge when three or more permutations are added**.

Figure 20 shows the corresponding training loss curves. We observe that when three or more permutations are applied, the model completely fails to converge. This aligns with the 3D shape visualizations in Figure 12, where the generated shapes do not represent any meaningful object.

# H  SYSTEM-LEVEL COMPARISON BETWEEN IMAGE AND WEIGHT GENERATION MODELS

In our study, we show that current generative models for weights primarily memorize training data, drawing a system-level comparison with the generalization of image generation models. Here, we provide a example to illustrate the generalization behaviors of a common image generation model trained on the same amount of data as weight generation models. Concretely, we compare HyperDiffusion, an unconditional diffusion model for weight generation, with a standard Denoising Diffusion Probabilistic Model (DDPM) (Ho et al., 2020) trained on Flowers (Nilsback & Zisserman, 2008).

We randomly select 2749 images from the 20 largest classes in Flowers, to match the dataset size used for HyperDiffusion. We then train an unconditional DDPM on this dataset, as well as on two smaller subsets of 100 and 500 images, applying horizontal flipping as data augmentation. All models are trained at a resolution of 64×64 with a consistent training setup: 43K iterations, batch size 64, 500 warm-up steps, and a learning rate of 1e-4.

| # training imgs | type | randomly sampled images |
|---|---|---|
| 100 | generated | |
|  | training | |
| 500 | generated | |
|  | training | |
| 2749 | generated | |
|  | training | |

Table 10: **Image diffusion models improve generalization and reduce memorization with more training data**. Each pair of consecutive rows shows randomly selected generated images alongside their most similar training images. When trained on 100 or 500 images, the model often replicates training samples or their horizontal flips—a data augmentation used during training. However, with 2749 training samples, the model generates novel images, demonstrating improved generalization.

After training the image diffusion models, we use the image copy detection method SSCD (Pizzi et al., 2022) to compute the similarity scores between generated and training images. Table 10 visualizes ten randomly selected generated images alongside their most similar training images. When trained on only 100 samples, the diffusion model primarily memorizes the training images, but with a larger dataset of 2749 images, it generalizes to produce novel outputs.

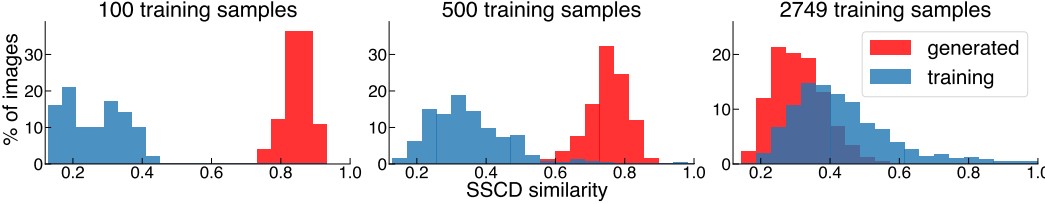

Figure 21: **Image diffusion models transition from memorization to generalization with more data**. The red histograms and blue curves show the distributions of SSCD similarity between each generated image and its most similar training image (red) and between each training image and its most similar training image (blue, excluding self-comparisons). As the training dataset grows, the red histograms shift left, indicating that the model generates increasingly distinct images rather than memorizing training samples.

Figure 21 presents the quantitative trend of similarity between generated images and their most similar training images as a function of dataset size. As a reference, we also show the similarity distribution between each training image and its most similar training image (excluding self-comparisons). We observe that with more data, the model generates images with a similarity level comparable to that between training images themselves. This *contrasts* with the trend observed for HyperDiffusion in Section 3.1, where the model fails to generate novel weights even when trained on 2749 samples.

**Discussion**. Here, we provide an example illustrating that, on a system-level, a common image generation models can generalize well on the same amount of data used to train weight generation models. However, this difference can be attributed to various factors, including but not limited to the generative model size and architecture, diversity of dataset, data modality, and training recipe.

# I  IMPACT OF DOWNSTREAM DATASET DIVERSITY ON WEIGHT MEMORIZATION

For the four methods studied in this paper, their primary experimental setups use SVHN, MNIST, ShapeNet, and CIFAR-100, respectively. One may wonder whether the limited diversity of these downstream datasets leads to limited diversity in the dataset of checkpoints, and thereby indirectly contributes to the memorization in the generative models of weights. To test this, we train p-diff on a more diverse image dataset, ImageNet (Deng et al., 2009), following its official codebase.

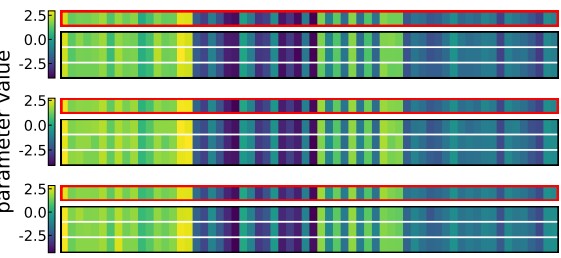 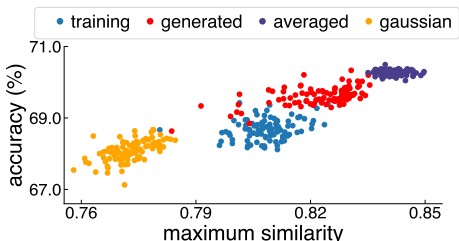

Figure 22: **P-diff's generated weights closely resemble training weights** when trained on ImageNet classification model checkpoints. The trend is consistent with the results on CIFAR-100 checkpoints (Figure 2).

Figure 23: **P-diff does not outperform interpolation baselines** when trained on ImageNet classification model checkpoints, similar to the CIFAR-100 results (Figure 6).

Similar to the results on CIFAR-100 checkpoints (Figures 2 and 6), the generated weight values remain highly similar to training weights, and fail to outperform the interpolation baselines in the accuracy-novelty trade-off, as shown in Figures 22 and 23. In addition, 84.4% of generated weight values fall within one standard deviation around the mean of the training values, compared to 68.7% for training weights themselves. This suggests that the generated ImageNet classification model weights also tend to concentrate around the mean of the training weights.

**Discussion**. While the diversity of the downstream image dataset may indirectly influence the generalization of generative models of weights, our results show that increasing image diversity alone does not reduce memorization. Instead, modeling choices and the diversity of checkpoint datasets (*e.g.*, training a generative model on only 300 checkpoints saved from a single run) may be more fundamental issues.

## J  INTRINSIC DIMENSION OF IMAGE AND WEIGHT DATA

### J.1  MAIN ANALYSIS

Unlike images, which are natural signals with high spatial redundancy, model weights have intricate dependencies between parameter groups. Weight data may exhibit higher complexity than images, potentially making it more challenging for generative models to capture their distribution, and limiting their ability to produce novel samples. To assess this complexity, we measure the *intrinsic dimensions* of weight and image data, which quantify the number of variables required to summarize high-dimensional data distributions.

**Estimation method**. We estimate the intrinsic dimensions using the Maximum Likelihood Estimator (MLE) (Levina & Bickel, 2004; MacKay & Ghahramani, 2005). It can characterize data beyond simple linear structure as identified in alternative methods such as the Principal Component Analysis (Pearson, 1901). In essence, it estimates intrinsic dimension by modeling neighbor distributions with a Poisson process and computing the maximum likelihood intrinsic dimension from observed distances to neighbors. Formally, the estimator is formulated as

$$\bar{m}_k = \left[ \frac{1}{n(k-1)} \sum_{i=1}^{n} \sum_{j=1}^{k-1} \log \frac{T_k(x_i)}{T_j(x_i)} \right]^{-1}, \tag{4}$$

where $\{x_i\}_{i=1}^{n}$ are the data points, $T_j(x_i)$ is the $L_2$ distance of $x_i$ to its $j$-th nearest neighbor, and $k$ is a hyperparameter that determines the number of nearest neighbors to consider.

MLE is shown to effectively capture the intrinsic dimensions of modern image datasets (Pope et al., 2021), but can be sensitive to the hyperparameter $k$ (the number of nearest neighbors considered in the estimation). Thus, we report estimations for $k = 3, 5, 10, 20$, following Pope et al. (2021).

**Data**. To compare the intrinsic dimensions of image and weight data, we use image datasets paired with the classification model weights trained on these datasets in Hyper-Representations.

Since the Maximum Likelihood Estimator requires the data to be independent and identically distributed (i.i.d.), we use only the weight checkpoint from the last epoch of each run to ensure that samples are i.i.d.. To align the image datasets we use with the datasets used to train the classification model checkpoints from Hyper-Representations, we resize all images to $28 \times 28$.

| dataset | $k=3$ | $k=5$ | $k=10$ | $k=20$ |
|---|---|---|---|---|
| MNIST (image) | 7 | 10 | 11 | 12 |
| MNIST (weight) | 56 | 79 | 86 | 85 |
| SVHN (image) | 8 | 13 | 16 | 17 |
| SVHN (weight) | 58 | 81 | 84 | 43 |
| CIFAR-10 (image) | 12 | 19 | 23 | 24 |
| CIFAR-10 (weight) | 62 | 89 | 99 | 100 |
| STL-10 (image) | 11 | 17 | 19 | 19 |
| STL-10 (weight) | 139 | 201 | 206 | 222 |

(a) raw data

| dataset | $k=3$ | $k=5$ | $k=10$ | $k=20$ |
|---|---|---|---|---|
| MNIST (image) | 10 | 14 | 17 | 18 |
| MNIST (weight) | 38 | 55 | 60 | 61 |
| SVHN (image) | 14 | 23 | 29 | 31 |
| SVHN (weight) | 45 | 55 | 49 | 37 |
| CIFAR-10 (image) | 19 | 33 | 42 | 44 |
| CIFAR-10 (weight) | 49 | 71 | 80 | 80 |
| STL-10 (image) | 21 | 33 | 37 | 36 |
| STL-10 (weight) | 58 | 81 | 89 | 88 |

(b) neural representations of data

Table 11: **MLE estimates weights to have higher intrinsic dimensions than images**, across different values of the hyperparameter $k$. We compute estimations for both raw data and their neural representations from an autoencoder. The estimations are rounded to integers.

**Images and weights**. Table 11a shows the intrinsic dimensions of image and weight datasets, measured with different values of hyperparameter $k$. We observe that, for all datasets and values of $k$, MLE consistently estimates much higher intrinsic dimensions for model weights than for images.

**Neural representations**. Aside from raw data, intrinsic dimension measures have also been used to inspect the neural representations of data (Ansuini et al., 2019; Yin et al., 2024). Here, we use

the estimators to quantify the intrinsic dimensions required for neural networks to capture the image and weight distributions. Concretely, we extract latent representations from autoencoders trained on model weights and images. For weight data, we use the pre-trained autoencoder from Hyper-Representations (Schürholt et al., 2021; 2022). For image data, we train a separate autoencoder with the same architecture, latent dimensions, and objectives.

Table 11b presents the MLE estimates for these latents. Consistent with our observation on raw data, the neural representations of weights have higher intrinsic dimensions than those of images. Interestingly, the neural representations of images have higher dimension estimates than raw images. This aligns with the "hunchback" pattern reported in prior work (Ansuini et al., 2019; Yin et al., 2024), where intrinsic dimension is low at the input layer due to dominant yet redundant features in images, but peaks in middle layers.

**Discussion**. Our results suggest that weight data have higher intrinsic dimensions than images, both in raw forms and neural representations. Although prior theoretical work has identified a negative relationship between the intrinsic dimensionality of data and the generalization of diffusion models (Chen et al., 2023; Oko et al., 2023), it is unclear whether the memorization in generative models of weights we observed is directly linked to the higher intrinsic dimensions of weight data.

## J.2 VALIDATING THE CONVERGENCE AND PERFORMANCE OF THE IMAGE AUTOENCODERS

In Appendix J.1, we trained autoencoders on image datasets to compare the intrinsic dimensions of the neural representations of image and weight data. Here, we verify the training of the image autoencoder by assessing its reconstruction quality in Table 12 and examining the reconstruction loss curves for test images in Figure 24. The results show that the autoencoder accurately reconstructs random test images, with the test loss stabilizing by the end of training.

| dataset | type | randomly sampled images |
|---------|------|-------------------------|
| MNIST | original | |
| | reconstructed | |
| SVHN | original | |
| | reconstructed | |
| CIFAR-10 | original | |
| | reconstructed | |
| STL-10 | original | |
| | reconstructed | |

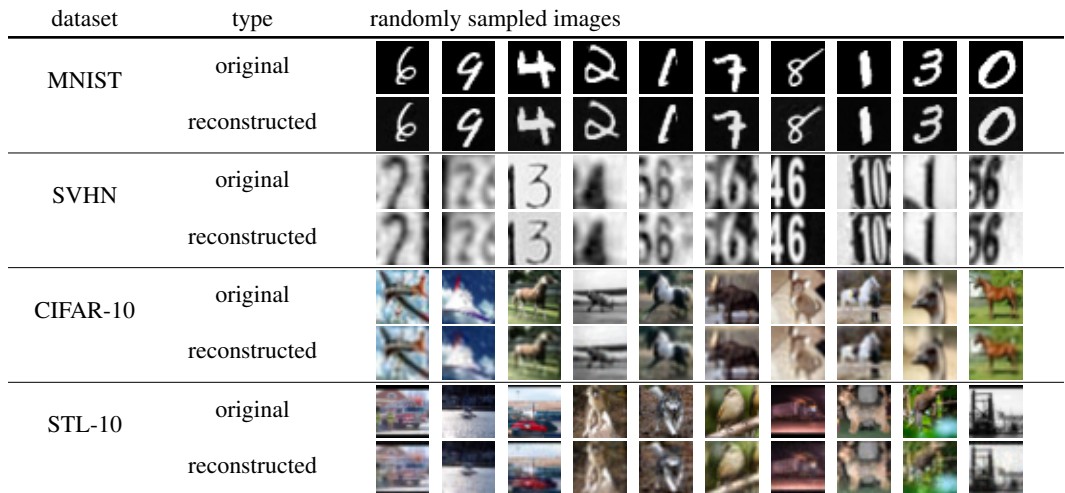

Table 12: **Reconstructions from the image autoencoders** in Appendix J.1.

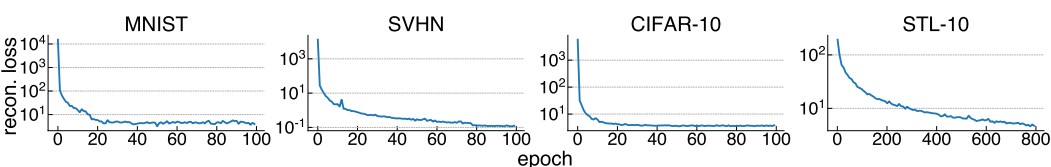

Figure 24: **Test loss curves for image autoencoder training** in Appendix J.1.

# K    CAN GENERATIVE MODELS BE USED TO STORE THE WEIGHT DATASETS?

Hegde et al. (2023) showed that diffusion models can be used to compress and store the weights of an archive of policy networks trained via Quality Diversity Reinforcement Learning (QD-RL), and enable flexible selection of specific behaviors from the policy archive. Since the generative models of weights we studied are primarily memorizing their training datasets, one might speculate whether this property could be used to compress and store the weight dataset in an alternative way.

To explore this possibility, we generate 20K checkpoints from HyperDiffusion and match each to its nearest training checkpoint. We note that only 129 (4.69%) out of 2749 training checkpoints are not replicated in generated checkpoints. Similarly, for G.pt, 4872 (47.63%) out of 10228 training checkpoints are not replicated in 50K generated checkpoints. These results suggest that generative models can indeed recover a substantial portion of the training weights.

However, the number of parameters in these generative models (223M for Hyper-Representations, 378M for G.pt, 1.4B for HyperDiffuion, and 9.6M for p-diff) far exceeds the total number of values in their respective training datasets (7.1M, 81M, 101M, and 0.6M). Therefore, storing the weight datasets implicitly within the generative models we studied would not be a space-efficient method.

## L    USAGE OF LARGE LANGUAGE MODELS IN PAPER WRITING

Large Language Models are used lightly during the final stage of paper writing to help shrink the main text to fit within 9 pages.

