# OpenReview forum: "Generative Modeling of Weights: Generalization or Memorization?"
_ICLR.cc/2026/Conference — ICLR 2026 Conference Withdrawn Submission_

### Official Review · Reviewer_wYhp · 2025-10-24

**Soundness:** 3
**Presentation:** 2
**Contribution:** 3
**Rating:** 6
**Confidence:** 4

**Summary:**

This paper rigorously examines the performance of contemporary generative models designed to synthesize neural network weights, specifically investigating whether these models achieve genuine generalization or primarily rely on memorization. Analyzing four representative methods—Hyper-Representations, G.pt, HyperDiffusion, and P-diff—the authors consistently find that generated checkpoints largely synthesize weights by memorization. The evidence for memorization is observed both in weight space, where generated weights are often replicas or simple interpolations of training checkpoints, and in model behavior, where generated models exhibit highly similar decision boundaries or reconstructed 3D shapes compared to their nearest training counterparts. Furthermore, these generative methods generally fail to outperform simple baselines such as adding Gaussian noise or using weight ensembles, in terms of generating diverse yet high-performing models. P-diff is identified as primarily functioning as a weight interpolator, producing models centered around the average of its low-diversity training distribution.

The authors attribute the pervasive memorization to several factors: limited training data, the highly overparameterized nature of the generative models, and the insufficient utilization of structural priors specific to weight data, such as permutation and scaling symmetries. The study highlights the need for more careful design and evaluation of generative models in this emerging domain.

**Strengths:**

1. Originality: The paper addresses a critical, fundamental question that challenges the prevailing narrative in the emerging area of generative modeling for neural network weights: whether high task performance actually indicates generalization or merely sophisticated data reproduction. This study is highly original as it moves beyond standard quality metrics (like downstream accuracy) to introduce and apply rigorous novelty and memorization metrics suitable for weight space data (e.g., comparing generated checkpoints to training checkpoints via L2 distance, error similarity, and utilizing the ZU score). It is the first comprehensive work to systematically test and debunk the claims of novelty made by prior work in this domain.

2. Quality: The paper demonstrates exceptional experimental quality and thoroughness. The analysis spans four distinct and representative generative modeling architectures (autoencoders, conditional diffusion, unconditional diffusion, latent diffusion) across different domains (classification models and neural field models). The quantitative results are compelling, showing generated checkpoints are significantly closer to training checkpoints than training checkpoints are to one another (Figure 4). The ablation studies investigating the roots of memorization are particularly strong: demonstrating that large models memorize random weights regardless of semantics (HyperDiffusion), and showing that scaling data can mitigate memorization for G.pt. The comparison against simple baselines (noise addition and interpolation) effectively contextualizes the observed performance, demonstrating that for most methods, the complexity of the generative model offers no clear advantage in novelty over simple heuristics.

3. Clarity: The paper is extremely clear and well-organized, making complex comparisons between models and mechanisms highly accessible. The dual approach of assessing novelty in both weight space (heatmaps, L2 distance) and behavioral space (decision boundaries, 3D shapes) provides a comprehensive view of the problem. The visualizations, such as the heatmaps in Figure 2 and the accuracy-novelty trade-off plots in Figure 6, provide intuitive and powerful support for the main arguments. The authors explicitly define the metrics used, including the modified similarity metric for classification models based on IoU of incorrect predictions.

4. Significance: The findings have broad significance for the future of generative modeling in non-traditional data modalities. By definitively demonstrating memorization across multiple leading approaches, the work sets a critical new standard for evaluation, emphasizing the necessity of testing generalization explicitly alongside performance. The identified causes—limited data, overparameterization, and failure to integrate structural priors (symmetries)—provide a crucial roadmap for future research, suggesting that architectural solutions invariant to weight symmetries may be necessary. The paper effectively shifts the focus from achieving high accuracy (which can be memorized) to achieving novel high-performing models.

**Weaknesses:**

I think the key weaknesses lie in the incomplete mechanistic understanding of why certain mitigations fail and the generalization constraints imposed by the training data sources themselves.

1. Inconsistent Results on Data Scaling: While the paper successfully demonstrates that scaling up the training data reduces memorization for G.pt (Table 2, Figure 9), it notes that scaling data does not mitigate memorization in Hyper-Representations (Table 7). This inconsistency, highlighted as potentially stemming from "more fundamental modeling issues", is a significant limitation on the generality of the proposed solution (scaling data). The paper requires a deeper investigation into why the intrinsic failure mode of Hyper-Representations resists data scaling, possibly linking it back to the autoencoder's inherent difficulty in accurately reconstructing training weights (Figure 3).

2. Limited Exploration of P-diff's Interpolation Mechanism: P-diff is explained as an interpolator because its training data (checkpoints saved at consecutive steps of a single training run) is inherently low-diversity. While the authors compare P-diff's outputs against simple interpolation baselines (averaging and Gaussian fitting) and find strong similarity, the conclusion that P-diff interpolates rather than generalizes is heavily constrained by the lack of diversity in its input data. The paper treats the consequence (interpolation) rather than the root cause of the model's inability to produce diverse weights when operating on a constrained manifold. The authors should formally discuss if the design of the P-diff method itself is fundamentally flawed for generalization when applied to this type of low-diversity, highly correlated training data.

3. Mechanistic Failure of Symmetry Augmentation: The results showing that permutation augmentation causes HyperDiffusion to fail to converge or generate meaningful shapes are extremely important, indicating that simple augmentation techniques are insufficient or even harmful for weight diffusion models. However, the paper only discusses architectural invariance as a necessary future step. A mechanistic analysis of why incorporating these function-preserving transformations destabilizes the training—e.g., how the high-dimensional permutation disrupts the diffusion dynamics or whether the resulting manifold is simply too complex to model—is missing and would greatly enhance the paper's contribution towards architectural redesign.

4. Limited Scope of Alternative Modeling Factors: Section 4.1 shows that overparameterization is a clear culprit (HyperDiffusion memorizes random weights). Section 4.3 explores various common modeling adjustments (reducing epochs, changing model size, regularization like dropout or Gaussian noise), finding that none significantly mitigate memorization without degrading performance. This suggests the problem is structural, but the paper primarily relies on simple modifications borrowed from image generation literature. More radical architectural changes or loss function design focusing explicitly on disentangling function-preserving variations (symmetries) from truly novel weight changes should be explored to provide stronger, actionable suggestions beyond general advice regarding graph neural networks.

**Questions:**

1. Data Scaling and Dataset Diversity:
- Question: The paper identifies limited data as a potential cause of memorization, demonstrating that scaling data helps G.pt (ZU score from -8.5 to 3.5). However, why did scaling the Hyper-Representations dataset (from 2,896 to 24,136 checkpoints) fail to reduce memorization (ZU score remained -13.6)? Was the nature or diversity of the scaled data for Hyper-Representations significantly different from G.pt's data? Were the additional checkpoints for Hyper-Representations derived from similar training runs or different runs? Clarifying this divergence is crucial for researchers determining whether they need to simply collect more data, or collect more functionally diverse data.
- Suggestion: Since training Hyper-Representations on scaled data still resulted in high memorization, could the authors perform an ablation where the generative model is trained specifically on the subset of training checkpoints that are maximally distant from each other in L2 space? This might test whether optimizing for training set diversity improves generalization, even if the total quantity of data remains limited.

2. Mechanistic Constraints on P-diff and Interpolation:
- Question: P-diff's training checkpoints are noted for their extremely low diversity, resulting from consecutive saving during a single training run. Given that P-diff appears to generate weights by interpolation within this narrow subspace, does this imply that diffusion models applied to the weight space are inherently constrained by the "local neighborhood" established by the optimization trajectory? If the input manifold is itself highly constrained, can the generative model ever truly generalize beyond it without architectural modifications?
- Question: P-diff trains an unconditional latent diffusion model. If the authors were to train P-diff using a checkpoint dataset known for high diversity (e.g., derived from independent runs, similar to G.pt's setup), would P-diff still exhibit the beneficial interpolation properties (higher accuracy than baselines) or would it collapse into the severe memorization patterns observed in Hyper-Representations and HyperDiffusion?

3. Understanding Symmetry Augmentation Instability:
- Question: The finding that even adding a single permutation to HyperDiffusion's training data leads to failure to produce meaningful shapes, and three or more permutations leads to failure of convergence, is a major finding regarding the difficulty of modeling weight distributions. Can the authors offer a more detailed hypothesis on the failure mechanism? Does permutation, being a drastic, discontinuous change in parameter values (even though function-preserving), create an optimization challenge for the diffusion model's noise prediction network that is specific to the highly rigid structure of the weight space data?
- Suggestion: The paper suggests architectural invariance (like Graph Neural Networks) as a path forward. A brief discussion or visualization comparing the high-dimensional geometry of the weight manifold after standard L2 perturbations versus after function-preserving permutation perturbations would help illustrate why these symmetries are so difficult to incorporate purely through data augmentation.

4. Novelty Metrics and Generated Quality Trade-off:
- Question: For HyperDiffusion, which generates neural field models, novelty is quantified by the minimum Chamfer Distance (CD) to training shapes (higher distance implies higher novelty). However, the visualized 100th percentile generated checkpoint shows degraded shape quality, suggesting it achieved 'novelty' primarily through low quality, rather than successful generalization (Figure 16c). How do the authors propose filtering or measuring successful novelty (i.e., high-performing models that are distant from the training set)? Should future work normalize the novelty metric by performance (e.g., only counting models above a specific performance threshold when calculating maximum similarity/minimum CD)? This is necessary to decouple successful generalization from simple noise-induced failure.
- Suggestion: Since model performance is crucial, it would be beneficial to report the correlation coefficients between the accuracy (or min CD to test set) and the novelty metrics (max similarity or min CD to training set) for the generated models. This could quantify the observed "accuracy-novelty trade-off" identified in Figure 6.

5. Role of Intrinsic Dimension:
- Question: The paper finds that model weight data exhibits higher intrinsic dimensions (ID) than image data. While this is hypothesized to increase the difficulty of generative modeling, the authors note the link is currently unclear. Could the authors perform a basic ablation study (if feasible within the current framework) correlating ID changes with memorization? For instance, does training G.pt on the 10x scaled dataset (which reduced memorization) correspond to a lower intrinsic dimension estimate compared to the original dataset? This would provide preliminary quantitative support for the ID hypothesis.

6. The Role of Generative Model Architecture in Memorization
The study emphasizes that generative models of weights are highly overparameterized (e.g., HyperDiffusion has 1.4B parameters for a small training set of 2749 neural fields). Furthermore, conventional techniques to mitigate memorization (reducing epochs, regularization, adjusting model size) failed to meaningfully improve novelty for G.pt, Hyper-Representations, and HyperDiffusion without degrading performance (Table 9).
- Question: Given that architectural changes and regularization failed to reduce memorization, does this imply that the standard U-Net/Transformer architecture common in diffusion models is fundamentally unsuited for capturing the symmetries and intrinsic structure of weight data? Could the authors elaborate on how the high capacity of the generative model specifically facilitates the observed exact replication of random weights in HyperDiffusion (ZU score of -30.8 across all random layer initializations)?
- Suggestion: The authors suggest architectural invariance (e.g., Graph Neural Networks) as a future solution. An intermediate step would be to investigate if applying explicit architectural constraints (e.g., parameter sharing or weight factorization common in hypernetworks) to the generative models could restrict their effective capacity enough to force generalization, even if performance initially drops.

**Details Of Ethics Concerns:**

No.

---

### Official Review · Reviewer_2ubX · 2025-10-31

**Soundness:** 2
**Presentation:** 3
**Contribution:** 2
**Rating:** 2
**Confidence:** 4

**Summary:**

This work proposes to study properties of weight prediction methods. Specifically, four main methods are studied: a hypernetwork-like method that produces a distribution over weights that can be sampled from and decoded into parameters, and three diffusion models for generating weights. Based on some simple metrics, we see that the method p.diff generates more diverse weights than other methods.
There is some further exploration of how each method performs when the data scale and model capacity is increased, as well as the effect of adding permutation information into the training process.

**Strengths:**

* This work is well written and easy to follow

* This is a very useful area of research, if we are able to produce the posterior distribution over neural network weights given some task or data, then many multi-agent or ensembling techniques will benefit. SO a critical study of what these approaches are really learning is important.

* The tests regarding generalization and memorization are sensible. Any method that purports to learn to generate weights should "pass" these tests.

**Weaknesses:**

* The conclusions are underwhelming. I would expect something (even general) prescription of what can be done to mitigate the issues here. From the appendix, it seems clear that the identified levers e.g. training data, model capacity, do not actually lay a role in improving the behavior of these methods.

* There is a mismatch in the amount of evidence presented for this study for each claim. Lots of space was dedicated to showing that the output weights are quite close to the training data. However, exploration of structural priors explored only amounts to adding some permutation information, which also does not work well. Scaling the model seems to have a similar deleterious effect. Taken together, I'm not sure what my conclusions should be beyond that these methods do not work well.

* This is a relatively simple study on a small number of methods with a small number of easy-to-compute metrics. To clear the bar for significance, I would expect a broader study of more methods, more data, metrics, etc.

**Questions:**

1) what can actually be done to mititgate issues with generating these (relatively) simple models? We are still talking about MNIST/CIFAR10 here.

---

### Official Review · Reviewer_2Hfk · 2025-10-31

**Soundness:** 2
**Presentation:** 4
**Contribution:** 2
**Rating:** 2
**Confidence:** 3

**Summary:**

The paper seeks to understand whether generative models of weights, such as p-diff, G.pt and others, generate novel model weights rather than memorizing their training sets. They do this by:
1) comparing models via the weights directly by measuring L2 distances.

2) comparing models via their outputs by measuring agreement on predictions.

The authors come to the conclusion that most models indeed memorize their training set rather than generate novel models.

**Strengths:**

1) The paper poses an important and interesting question regarding the capabilities of modern generative models, specifically in the domain of weight generation.

2) The paper is convincing in showing that generated samples of weights are very similar to those found in the training set.

3) The paper is clearly written and easy to follow.

**Weaknesses:**

My main concern revolves around defining and measuring novelty, the key issue which this paper focuses on. The only definition I found in the paper is in the abstract - "weights that are different from the checkpoints seen during training", seems not to fall in line with other details in the paper (more on that under points (2)). I think that the paper would extremely benefit from a clear definition of novelty, and a somber look at if this should be expected from these weight generation models, given their training settings. If indeed the authors agree that under these settings memorization is inevitable, the paper could benefit from shifting the focus to experiments such as in section 4, and maybe suggesting more solutions.

I detail this concern below.

**Major Weaknesses**

1) The concept of “Novel Weights” - In my opinion, the main weakness of this paper is the definition of novelty. Each generative model examined is trained on multiple training runs, and from each run multiple checkpoints are used. What do you expect novel weights look like under this setting? Are they similar to training models from scratch using a different random seed? Are novel weights simply any weights not in the training set that achieves a low error rate?

    Additionally, under this setting wouldn’t it be reasonable to expect the generative models to simply generate new checkpoints for the training runs that they’ve seen during training?

    Finally, a strict definition of ‘novelty’ is required to claim that interpolation is not generalization (section 3.3).


2) Measuring $\ell_2$ - I agree that it is a good way to show that a high degree of similarity between weights, but I would argue that this isn't a good measure for 'novelty'. Are two sets of weights which are permutations of each other considered novel? They will have a large $\ell_2$distance as you rightfully note. Another example where this is problematic - in figure 6 you show that weights with added gaussian noise perform similarly to the 'clean' weights, but these new weights will have large $\ell_2$ distance from the 'clean' weights as the number of parameters increases.


3) IoU metric - The intersection over union method seems similar to previous methods of comparing neural networks [1] which find that different neural networks tend to agree on which images are classified correctly\incorrectly (but don’t necessarily agree on the incorrect label). So maybe we should expect high similarity in this case even for ‘novel’ models?


**Minor Weaknesses**

1) Figure 4 is a bit misleading. Since the models are trained with multiple checkpoints of the same training run, one could assume that a new sample from the generative model would simply be a new checkpoint, therefore have a small L2 distance to training examples. However, you do not take this into account for the blue histograms, as explained in lines 200-201 and in appendix A.

2) While figure 5 is indeed surprising and unlikely, I am not sure what we could take away from this - to my understanding, the PCA figures are generated using images consisting of the first 2 PCAs of natural images? This would mean that the figure is generated using flat colored images rather than natural ones. Here I would like to see the output of a random initialization as well for reference.
Interpolation isn’t generalization - again since the notion of generalization \ novelty isn’t defined, I’m not sure that you’re right to say that interpolation isn’t generalization.

3) Figure 6.b,c - the scatter of training and generated points looks very similar, isn’t this what you would like your generative model to look like?



**Suggestions**

1) A better definition of novelty in this context. This definition should probably take into account the fact that the training examples are all neural networks trained with gradient descent, rather than all possible functions realized by neural networks.

    If a new checkpoint for a training session found in the training data is not considered ‘novel’, maybe you should consider retraining the models without multiple checkpoints?

2) Lines 302-304: To measure the ‘benefits’ of generation methods, you can try scatter plotting the accuracy of generated models vs the accuracy achieved by the most similar training point.

3) Better baselines - In the case of figure 4, where is a permuted network on the x-axis? Where is a random initialization? Where are networks with added gaussian noise? In the case of figure 6, where are models from different architectures from the one you are generating.



[1] https://arxiv.org/pdf/2102.03349

**Questions:**

1) Isn’t a trivial explanation for ‘memorization’ the fact that most of these networks simply have less training examples than the input dimension? Regarding results on HyperDiffusion - from my understanding they train their generative model on ~14K different shape representations (section 6 in their paper), but each training example is of dimension ~36K (section 5 in their paper).

    This seems to be the same case in Hyper-representation where they train on ~3K checkpoints of CNNs, each consisting of ~3K weights (table 5 in appendix A). Since many checkpoints are correlated with each other (they are subsequent checkpoints from the same training run) the effective size of the training set is smaller than the number of parameters.


2) Table 1 - How exactly did you generate this table? Did you sample N generated networks and then compare them to the training set? Does this mean that some of the networks sampled are simply random initializations?

---

### Official Review · Reviewer_vFBH · 2025-11-07

**Soundness:** 3
**Presentation:** 3
**Contribution:** 3
**Rating:** 6
**Confidence:** 5

**Summary:**

This paper systematically investigates whether recent approaches in generative modeling of neural network weights truly generalize or merely memorize training checkpoints. By analyzing several representative methods, the authors demonstrate that most models reproduce or interpolate between known checkpoints rather than generating genuinely novel weights. Empirical comparisons reveal that current generative weight models exhibit limited generalization capacity. The study attributes this to the narrow diversity of training checkpoints, the memorization ability of over-parameterized generators, and the lack of structural priors reflecting weight-space symmetries. Overall, the paper concludes that generative weight modeling remains largely a memorization process and calls for deeper theoretical and empirical understanding of the geometry of weight space.

**Strengths:**

- The paper proposes and empirically validates three hypotheses regarding memorization phenomena in the generative modeling of weights.
- It provides clear quantitative indicators to measure whether a generative model reproduces training samples by memorization or by genuine generalization.
- The analysis sheds light on the underlying mechanisms of model weight generation, offering insights for understanding and evaluating future generative approaches in parameter space.

**Weaknesses:**

- The noise-added baseline experiments are somewhat questionable, as recent findings do not sufficiently justify the smoothness of model weight space. Adding random noise in weight space may not correspond to meaningful perturbations in function space.
- The related work section (2.2) lacks a comprehensive discussion of literature on symmetry and invariance in model weight space. Since the paper focuses on analyzing the geometry and structure of weight distributions rather than proposing a new method, stronger grounding in related theoretical works would improve credibility and context.

**Questions:**

**Validity of noise-based baselines**
- The use of random noise perturbation in weight space is problematic because the structure of weight space remains poorly understood. To meaningfully test robustness or smoothness, perturbations should be performed in data or latent space rather than directly in weight space.
- For example, follow-up works on HyperNetworks (e.g., GEM, GASP, Functa, mNIF) demonstrate that weight generation operates over a latent vector space with learned smoothness. In contrast, direct parameter perturbation lacks theoretical grounding.
  - [GEM] Learning Signal-Agnostic Manifolds of Neural Fields, NeurIPS 2021
  - [GASP] Generative Models as Distributions of Functions, AISTATS 2022
  - [Functa] From Data to Functa: Your Data Point is a Function and You Can Treat It Like One, ICML 2022
  - [mNIF] Generative Neural Fields by Mixtures of Neural Implicit Functions, NeurIPS 2023
- A more principled analysis might involve perturbing input data (e.g., via data augmentation) and examining variance between resulting trained parameters. If parameter variance under strong augmentation does not differ significantly from overall variance across the dataset, this could support the memorization hypothesis.
- In addition, Git Re-Basin and DWSNet explicitly discuss permutation invariance and equivariance in parameter space, which could inform how meaningful addition or averaging operations should be performed.
  - [Git Re-Basin] Merging Models modulo Permutation Symmetries, ICLR 2023
  - [DWSNet] Equivariant Architectures for Learning in Deep Weight Spaces, ICML 2023

**Regarding HyperDiffusion experiments**
- The authors themselves note that HyperDiffusion only works when models share the same initialization. Therefore, experiments such as Figure 12 may not be valid unless all models are initialized identically.
- A more appropriate approach would be to perturb training samples in data space to increase sample diversity, as this maintains meaningful parameter correlations. This would lead to more reliable conclusions regarding generalization vs. memorization.

**Extension to generative models of data**
- Beyond weight generation, could a similar memorization analysis be extended to data-space generative models (e.g., diffusion or autoregressive models)? While these models appear to generalize due to interpolation and sample diversity, it remains unclear whether they actually memorize training examples. Incorporating relevant prior work that investigates memorization in data generative models would strengthen the discussion.

---

### Note · Authors · 2025-11-14

I have read and agree with the venue's withdrawal policy on behalf of myself and my co-authors.